# Steady state particle swarm

Carlos M. Fernandes[1,*], Nuno Fachada[1,2,*], Juan-Julián Merelo[3] and Agostinho C. Rosa[1]

[1] LARSyS: Laboratory for Robotics and Systems in Engineering and Science, University of Lisbon, Lisbon, Portugal
[2] HEI-Lab—Digital Human-Environment and Interactions Lab, Universidade Lusófona de Humanidades e Tecnologias, Lisbon, Portugal
[3] Department of Architecture and Computer Technology, University of Granada, Granada, Spain
* These authors contributed equally to this work.



## ABSTRACT

This paper investigates the performance and scalability of a new update strategy for the particle swarm optimization (PSO) algorithm. The strategy is inspired by the Bak–Sneppen model of co-evolution between interacting species, which is basically a network of fitness values (representing species) that change over time according to a simple rule: the least fit species and its neighbors are iteratively replaced with random values. Following these guidelines, a steady state and dynamic update strategy for PSO algorithms is proposed: only the least fit particle and its neighbors are updated and evaluated in each time-step; the remaining particles maintain the same position and fitness, unless they meet the update criterion. The steady state PSO was tested on a set of unimodal, multimodal, noisy and rotated benchmark functions, significantly improving the quality of results and convergence speed of the standard PSOs and more sophisticated PSOs with dynamic parameters and neighborhood. A sensitivity analysis of the parameters confirms the performance enhancement with different parameter settings and scalability tests show that the algorithm behavior is consistent throughout a substantial range of solution vector dimensions.

## INTRODUCTION

Particle swarm optimization (PSO) is a social intelligence model for optimization and learning (*Kennedy & Eberhart, 1995*) that uses a set of position vectors (or *particles*) to represent candidate solutions to a specific problem. Every particle is evaluated by computing its fitness, after its speed and position are updated according to local and global information about the search. During the search, the particles move through the fitness landscape of the problem, following a simple set of equations that define the velocity (Eq. (1)) and position (Eq. (2)) of each particle in each time step and drive them heuristically toward optimal regions of a $D$-dimensional search space. Here, Eqs. (1) and (2) describe a variant proposed by *Shi & Eberhart (1999)* that is widely used in PSO implementations. The difference to the original PSO is the introduction of the inertia weight parameter ω in order to help (together with $c_1$ and $c_2$) fine-tuning the balance between local and global search. All PSO implementations in this paper use inertia weight.

Corresponding author
Carlos M. Fernandes,
cfernandes@laseeb.org

The velocity $v_{i,d}$ and position $x_{i,d}$ of the $d$-th dimension of the $i$-th particle are therefore updated as follows:

$$v_{i,d}(t) = \omega v_{i,d}(t-1) + c_1 r1_{i,d}\left(pbest_{i,d} - x_{i,d}(t-1)\right) + c_2 r2_{i,d}\left(gbest_{i,d} - x_{i,d}(t-1)\right) \quad (1)$$

$$x_{i,d}(t) = x_{i,d}(t-1) + v_{i,d}(t) \quad (2)$$

where $\vec{X}_i = (x_{i,1}, x_{i,2}, \dots x_{1,D})$ is the position vector of particle $i$; $\vec{V}_i = (v_{i,1}, v_{i,2}, \dots v_{1,D})$ is the velocity of particle $i$; $\vec{pbest}_i = (pbest_{i,1}, pbest_{i,2}, \dots pbest_{1,D})$ is the best solution found so far by particle $i$; $\vec{gbest}_i = (gbest_{i,1}, gbest_{i,2}, \dots gbest_{1,D})$ is the best solution found so far by the neighborhood of particle $i$. The neighborhood of a particle is defined by the network configuration that connects the population and structures the information flow. Parameters $r1_{i,d}$ and $r2_{i,d}$ are random numbers uniformly distributed within the range $(0, 1)$ and $c_1$ and $c_2$ are the acceleration coefficients, which are used to tune the relative influence of each term of the formula.

Most of the PSOs use one of two simple sociometric principles for constructing the neighborhood network (which defines the $\vec{g}$best values). *G*best (where *g* stands for global) connects all the members of the swarm to one another. The degree of connectivity of *g*best is $k = n$, where *n* is the number of particles. *L*best (where *l* stands for local), creates a neighborhood with the particle itself and its *k* nearest neighbors. A particular case of the *l*best topology is the *ring* structure, in which the particles are arranged in a ring, with a degree of connectivity $k = 3$, including the particle itself. Between the $k = 3$ connectivity of *l*best ring and $k = n$ of *g*best, there are several possibilities. Two of the most used are the two-dimensional square lattices with von Neumann and Moore neighborhoods.

Usually, PSOs are synchronous, meaning that first, the fitness values of all vectors must be computed, and only then their velocity is updated. However, there is another possible approach, in which the velocity of the particles is updated immediately after computing the fitness. In this case, the particles move with incomplete knowledge about the global search: if, for instance, the underlying network connecting the particles is a regular graph, then, on average, each particle is updated knowing the current best position found by half of its neighbors and the previous best found by the other half. This variant, which is called asynchronous PSO (A-PSO), was tested by *Carlisle & Dozier (2001)*. In the paper, the authors claim that A-PSO yields better results than the synchronous version (i.e., S-PSO), but since then other authors reached different conclusions: *Engelbrecht (2013)* and *Rada-Vilela, Zhang & Seah (2013)*, for instance, reported that S-PSO is better than A-PSO in terms of the quality of the solutions and convergence speed.

The importance of investigating update strategies for PSO lies in the possibility of distributed computation (*McNabb, 2014*). Even though standard PSOs can be easily parallelized—a particle or a set of particles can be assigned to each processor, for instance—, load imbalances may cause an inefficient use of the computational resources if synchronous updates are used. Asynchronous strategies do not require that all particles in the population have perfect knowledge about the search before the update step (a requirement that may cause idle processor times in a synchronous implementation), and therefore are a valid approach for parallelizing particle swarms. In addition,

asynchronism can also be useful in preventing premature convergence (*Aziz et al., 2014*), or to speed up convergence by skipping function evaluations (*Majercik, 2013*).

Here, we are mainly concerned with performance issues, in general, and convergence speed in particular. The goal is to design an A-PSO that, unlike the standard A-PSO, significantly improves on the convergence speed of S-PSO in a wide range of problems. We hypothesize that reducing the number of evaluations in each time step, while focusing only on harder cases (i.e., worst solutions), reduces the total number of evaluations required to converge to a specific criterion, that is, the computational effort to reach a solution. With that objective in mind, we have designed and implemented a novel strategy for one of the fundamental mechanisms of PSO: the velocity update strategy. Following the nature of the method, the algorithm has been entitled steady state PSO (SS-PSO).

In systems theory, a system is said to be in steady state when some of its parts do not change for a period of time (*Baillieul & Samad, 2015*). SS-PSO only updates and evaluates a fraction of the population in each time step: the worst particles and its neighbors, thus imposing a kind of selection pressure upon the whole population. The other particles remain in the same position until they eventually fulfill the criterion (being the worst particle or one of its neighbors).

Steady state replacement strategies are common in other population-based metaheuristics, namely Evolutionary Algorithms (*Whitley & Kauth, 1988*). However, steady state populations are much less frequent in PSO (*Majercik, 2013*; *Fernandes et al., 2014*; *Allmendiger, Li & Branke, 2008*). In fact, the strategy proposed in this paper is, to the extent of the authors' knowledge, the first that uses dynamic steady state update coupled with selective pressure. Furthermore, results demonstrate that the criterion for selecting the pool of individuals to update is very important for the success of the update strategy: the update step should be restricted to the worst individuals and their neighbors for optimizing performance. With this design, the steady state update strategy is not only able to improve the convergence speed of PSO standard configurations, but also more sophisticated variants of the algorithm, such as PSOs with time-varying parameters (*Ratnaweera, Halgamuge & Watson, 2004*) and dynamic neighborhood (*Vora & Mirlanalinee, 2017*).

The strategy was inspired by the Bak–Sneppen model of co-evolution between interacting species and by the theory of self-organized criticality (SOC) (*Bak & Sneppen, 1993*). SOC is a property of some systems that have a critical point as an attractor. However, unlike classical phase transitions, where a parameter needs to be tuned for the system to reach critical point, SOC systems spontaneously reach that critical state between order and randomness. In a SOC system near the critical point, small disturbances can cause changes of all magnitudes. These events, which are spatially or temporally spread through the system, are known as *avalanches*.

Avalanches occur independently of the initial state. Moreover, the same perturbation may cause small or large avalanches, depending on the current state of the system—that is, its proximity to the critical point. The distribution of avalanches during a large period displays a power-law between their size and frequency: small avalanches occur very often while large events that reconfigure almost the entire system are scarcer. SOC complex

systems balance between stability and creative destruction. In fact, power-law relationships between the size of events and their frequency, one of SOC's signatures, are widespread in Nature. Earthquake distribution, for instance, follows the Gutenberg-Richter law (*Gutenberg & Richter, 1956*), a power-law proportion between the magnitude of the earthquakes that occurred in a specific area during a specific period of time, and the frequency of those earthquakes.

Self-organized criticality was studied for the first time in the sandpile model (*Bak, Tang & Wiesenfeld, 1987*). Since then, the concept has been extended to other complex systems: besides the aforementioned earthquakes, the proponents of the theory claim that SOC may be a link between a broad range of phenomena, like forest-fires, ecosystems, financial markets and the brain (*Bak, 1996*). One of such systems is the Bak–Sneppen model of co-evolution between interacting species (*Bak & Sneppen, 1993*).

The Bak–Sneppen model was developed with the main objective of trying to understand the mechanisms underlying mass extinctions in nature. Ecosystems are complex adaptive systems in which the agents (the natural species) are related through several features, like food chains or symbiosis, for instance. In such interconnected environments, the extinction of one species affects the species that are related to it, in a chain reaction that can be of any size: in fact, fossil records suggest that the size of extinction outbreaks is in power-law proportion to their frequency.

In order to model the extinction patterns in nature and search for SOC signatures in co-evolutionary systems, *Bak & Sneppen (1993)* structured a set of species in a ring network and assigned a fitness value to each. Then, in every time step, the least fit species and its neighbors are eliminated from the system and replaced by individuals with random fitness. To put it in mathematical terms, the system is defined by $n$ fitness values arranged as a ring (ecosystem). At each time step, the smallest value and its two neighbours are replaced by uncorrelated random values drawn from a uniform distribution. Operating with this set of rules, the system is driven to a critical state where most species have reached a fitness value above a certain threshold. Near the critical point, extinction events of all scales can be observed.

Self-organized criticality theory has been a source of inspiration for metaheuristics and unconventional computing techniques. Extremal optimization (EO) (*Boettcher & Percus, 2003*), for example, is based in the Bak–Sneppen model. EO uses a single solution vector that is modified by local search. The algorithm removes the worst components of the vector and replaces them with randomly generated material. By plotting the fitness of the solution, it is possible to observe distinct stages of evolution, where improvement is disturbed by brief periods of dramatic decrease in the quality.

*Løvbjerg & Krink (2002)* modeled SOC in a PSO in order to control the convergence of the algorithm and maintain population diversity. The authors claim that their method is faster and attains better solutions than the standard PSO. However, the algorithm adds several parameters to the standard PSO parameter set: overall five parameters must be tuned or set to constant ad hoc values.

Complex and dynamic population structures have been one of most popular PSO research areas in the last decade. The comprehensive-learning PSO (CLPSO) (*Liang et al., 2006*;

*Lynn & Suganthan, 2015*) abandons the global best information, replacing it by a complex and dynamic scheme that uses all other particles' past best information. The algorithm significantly improves the performance of other PSOs on multimodal problems.

*Ratnaweera, Halgamuge & Watson (2004)* propose new parameter automation strategies that act upon several working mechanisms of the algorithm. The authors introduce the concepts of time-varying acceleration coefficients (PSO-TVAC) and also *mutation*, by adding perturbations to randomly selected modulus of the velocity vector. Finally, the authors describe a *self-organizing hierarchical particle swarm optimizer with time-varying acceleration coefficients*, which restricts the velocity update policy to the influence of the cognitive and social part, reinitializing the particles whenever they are stagnated in the search space.

*Liu, Du & Wang (2014)* describe a PSO that uses a scale-free (SF) network for connecting the individuals. SF-PSO attains a better balance between solution quality and convergence speed when compared to standard PSOs with *g*best and *l*best neighborhood topology. However, the algorithm is not compared under more sophisticated frameworks or against state-of-the art PSOs. Furthermore, the size of the test set is small and does not comprise shifted or rotated functions.

Finally, *Vora & Mirlanalinee (2017)* propose a dynamic small world PSO (DSWPSO). Each particle communicates with the four individuals of its von Neumann neighborhood, to which two random connections are added (and then removed) in each time step. In other words, the neighborhood of each particle is comprised of six particles, four of them fixed throughout the run while the remaining two keep changing. The authors compare the performance of DSWPSO with other PSOs and conclude that due to a more balanced exploration and exploitation trade-off, DSWPSO is consistently better.

In this work, the Bak–Sneppen model is used to design an alternative update strategy for the PSO. The strategy has been previously tested on a set of benchmark functions and compared to a standard S-PSO (*Fernandes, Merelo & Rosa, 2016*). The results show that SS-PSO significantly improves the performance of a S-PSO structured in a two-dimensional square lattice with Moore neighborhood. This paper is an extension of the aforementioned work. The main contributions here are: (a) a complete statistical analysis of the performance, comparing the algorithm with standard PSOs and variations of the proposed strategy; (b) a parameter sensitivity analysis and scalability tests showing that the performance enhancement introduced by the steady-state strategy is maintained throughout a reasonable range of parameter values and search space dimension ranging from 10 to 50; and (c) a comparison with state-of-the-art dynamic PSOs: CLPSO, PSO-TVAC and DSWPSO.

# MATERIALS AND METHODS

## SS-PSO algorithm

Steady state PSO was inspired by a similarity between PSO and the Bak–Sneppen model: both are population models in which the individuals are structured by a network and evolve toward better fitness values. With this likeness in mind, we have devised an

---

**Algorithm 1** Steady state particle swarm optimization.

for all particles $i \in \{1, 2 \ldots \mu\}$ do

    initialize velocity and position of particle $i$

    compute fitness of particle $i$

end

for all particles $i \in \{1, 2 \ldots \mu\}$ do

    compute $p$best and $g$best of particle $i$

end

repeat

    update velocity (Eq. (1)) of particle with worst fitness and its neighbors

    update position (Eq. (2)) of particle with worst fitness and its neighbors

    compute fitness of particle with worst fitness and its neighbors

    for all particles $i \in \{1, 2 \ldots \mu\}$ do

        compute $p$best and $g$best of particle $i$

until *termination criterion is met*

---

asynchronous and steady state update strategy for PSO in which only the least fit particle and its neighbors are updated and evaluated in each time step. Please note that SS-PSO is not an extinction model like the Bak–Sneppen system: the worst particle and its neighbors are not replaced by random values; they are updated according to Eqs. (1) and (2). As for the other particles, they remain steady—hence the name of the algorithm: SS-PSO.

The particles to be updated are defined by the social structure. For instance, if the particles are connected by a *l*best topology with $k = 3$, then only the worst particle and its two nearest neighbors are updated and evaluated. Please note that local synchronicity is used here: the fitness values of the worst and its neighbors are first computed and only then the particles update their velocity. For the remaining mechanisms and parameters, the algorithm is exactly as a standard PSO. For a detailed description of SS-PSO, please refer to Algorithm 1.

The PSOs discussed in this paper, including the proposed SS-PSO, are available in the OpenPSO package, which offers an efficient, modular and multicore-aware framework for experimenting with different approaches. OpenPSO is composed of three modules:

1. A PSO algorithm library.
2. A library of benchmarking functions.
3. A command-line tool for directly experimenting with the different PSO algorithms and benchmarking functions.

The library components can be interfaced with other programs and programming languages, making OpenPSO a flexible and adaptable framework for PSO research. Its source code is available at https://github.com/laseeb/openpso.

**Table 1 Benchmark functions.**

| | Mathematical representation | Range of search/initialization | Stop criterion |
|---|---|---|---|
| Sphere $f_1$ | $f_1(\vec{x}) = \sum_{i=1}^{D} x_i^2$ | $(-100, 100)^D$ $(50, 100)^D$ | 0.01 |
| Quadric $f_2$ | $f_2(\vec{x}) = \sum_{i=1}^{D} \left( \sum_{j=1}^{i} x_j \right)^2$ | $(-100, 100)^D$ $(50, 100)^D$ | 0.01 |
| Hyper Ellipsoid $f_3$ | $f_1(\vec{x}) = \sum_{i=1}^{D} i x_i^2$ | $(-100, 100)^D$ $(50, 100)^D$ | 0.01 |
| Rastrigin $f_4$ | $f_4(\vec{x}) = \sum_{i=1}^{D} (x_i^2 - 10\cos(2\pi x_i) + 10)$ | $(-10, 10)^D$ $(2.56, 5.12)^D$ | 100 |
| Griewank $f_5$ | $f_5(\vec{x}) = 1 + \frac{1}{4,000} \sum_{i=1}^{D} x_i^2 - \prod_{i=1}^{D} \cos\left(\frac{x_i}{\sqrt{i}}\right)$ | $(-600, 600)^D$ $(300, 600)^D$ | 0.05 |
| Schaffer $f_6$ | $f_6(\vec{x}) = 0.5 + \frac{\left(\sin\sqrt{x^2+y^2}\right)^2 - 0.5}{(1.0 + 0.001(x^2+y^2))^2}$ | $(-100, 100)^2$ $(15, 30)^2$ | 0.00001 |
| Weierstrass $f_7$ | $f_7(\vec{x}) = \sum_{i=1}^{D} \left( \sum_{k=0}^{k_{max}} \left[ a^k \cos(2\pi b^k (x_i + 0.5)) \right] \right) - D \sum_{k=0}^{k_{max}} \left[ a^k \cos(2\pi b^k \cdot 0.5) \right],$ $a = 0.5, b = 3, k_{max} = 20$ | $(-0.5, 0.5)^D$ $(-0.5, 0.2)^D$ | 0.01 |
| Ackley $f_8$ | $f_8(\vec{x}) = -20\exp\left(-0.2\sqrt{\frac{1}{D}\sum_{i=1}^{D} x_i^2}\right) - \exp\left(\frac{1}{D}\sum_{i=1}^{D}\cos(2\pi x_i)\right) + 20 + e$ | $(-32.768, 32.768)^D$ $(2.56, 5.12)^D$ | 0.01 |
| Shifted Quadric with noise $f_9$ | $f_9(\vec{z}) = \sum_{i=1}^{D} \left( \sum_{j=1}^{i} z_j \right)^2 * (1 + 0.4|N(0,1)|),$ $\vec{z} = \vec{x} - \vec{o}, \vec{o} = [o_1, ..o_D]$: shifted global optimum | $(-100, 100)^D$ $(50, 100)^D$ | 0.01 |
| Rotated Griewank $f_{10}$ | $f_{10}(\vec{z}) = 1 + \frac{1}{4,000} \sum_{i=1}^{D} z_i^2 - \prod_{i=1}^{D} \cos\left(\frac{z_i}{\sqrt{i}}\right), \vec{z} = M\vec{x}$, M: orthogonal matrix | $(-600, 600)^D$ $(300, 600)^D$ | 0.05 |

## Experimental setup

For testing the algorithm, 10 benchmark problems (Table 1) are used. Functions $f_1$–$f_3$ are unimodal; $f_4$–$f_8$ are multimodal; $f_9$ is the shifted $f_2$ with noise and $f_{10}$ is the rotated $f_5$ ($f_9$ global optimum and $f_{10}$ matrix were taken from the CEC2005 benchmark). Population size μ is set to 49. This particular value, which lies within the typical range (*Kennedy & Eberhart, 1995*), was set in order to construct square lattices with von Neumann and Moore neighborhood. Following (*Rada-Vilela, Zhang & Seah, 2013*), $c_1$ and $c_2$ were set to 1.494 and ω to 0.7298. $X_{max}$, the maximum position value, and $V_{max}$, the maximum velocity value, are defined by the domain's upper limit. Asymmetrical initialization is used, with the initialization ranges in Table 1. Each algorithm was executed 50 times with each function and statistical measures were taken over those 50 runs. Stop criteria have been defined according to the functions and objectives of the experiments (see details in the section "Results").

This work reports an extensive study of the proposed methodology. Different kinds of experiments have been performed, each one to investigate different aspects of the steady-state update strategy. The first experiment attempts at a proof-of-concept: SS-PSO

**Table 2  Median, minimum and maximum best fitness (50 runs).**

| | S-PSO$_{lbest}$ | | | S-PSO$_{VN}$ | | | S-PSO$_{Moore}$ | | |
|---|---|---|---|---|---|---|---|---|---|
| | Median | Min | Max | Median | Min | Max | Median | Min | Max |
| $f_1$ | 4.57$e$−06 | 9.44$e$−07 | 2.83$e$−05 | 9.13$e$−10 | 1.68$e$−10 | 6.70$e$−09 | **5.05$e$−12** | 8.81$e$−13 | 4.43$e$−11 |
| $f_2$ | 5.39$e$−13 | 3.09$e$−15 | 1.57$e$−11 | 4.52$e$−23 | 3.06$e$−25 | 2.81$e$−21 | **1.18$e$−30** | 1.01$e$−33 | 9.41$e$−28 |
| $f_3$ | 3.01$e$−05 | 8.44$e$−06 | 1.65$e$−04 | 5.58$e$−09 | 1.16$e$−09 | 4.60$e$−08 | **2.53$e$−11** | 3.08$e$−12 | 1.94$e$−10 |
| $f_4$ | 1.09$e$+02 | 6.57$e$+01 | 1.53$e$+02 | **6.02$e$+01** | 3.38$e$+01 | 1.09$e$+02 | 5.17$e$+01 | 3.78$e$+01 | 1.13$e$+02 |
| $f_5$ | **0.00$e$00** | 0.00$e$00 | 7.40$e$−03 | **0.00$e$00** | 0.00$e$00 | 5.38$e$−02 | **0.00$e$00** | 0.00$e$00 | 4.92$e$−02 |
| $f_6$ | **0.00$e$00** | 0.00$e$00 | 9.72$e$−03 | **0.00$e$00** | 0.00$e$00 | 0.00$e$00 | **0.00$e$00** | 0.00$e$00 | 9.72$e$−03 |
| $f_7$ | **0.00$e$00** | 0.00$e$00 | 0.00$e$00 | **0.00$e$00** | 0.00$e$00 | 3.29$e$−02 | 9.03$e$−04 | 0.00$e$00 | 1.12$e$00 |
| $f_8$ | 1.33$e$−15 | 8.88$e$−16 | 1.33$e$−15 | 1.33$e$−15 | 8.88$e$−16 | 1.33$e$−15 | **8.88$e$−16** | 8.88$e$−16 | 1.33$e$−15 |
| $f_9$ | 1.74$e$+02 | 3.41$e$+01 | 1.07$e$+03 | 4.76$e$−02 | 4.87$e$−04 | 2.05$e$+02 | **9.80$e$−05** | 6.44$e$−07 | 1.64$e$+03 |
| $f_{10}$ | **0.00$e$00** | 0.00$e$00 | 9.86$e$−03 | **0.00$e$00** | 0.00$e$00 | 3.19$e$−02 | 7.40$e$−03 | 0.00$e$00 | 5.19$e$−01 |

**Note:**
Best median fitness among the three algorithms shown in bold.

is compared with standard (and synchronous) update strategies. The objective of the second experiment is to check if the convergence speed-up is caused indeed by the selective strategy or instead by the restricted evaluation pool, which is a consequence of the proposed method. The third test aims at studying the parameter sensitivity and the scalability with problem size. For that purpose, several tests have been conducted in a wide range of parameter values and problem dimension. The fourth experiment investigates SS-PSO under time-varying parameters and experiment number five compares SS-PSO with dynamically structured PSOs.

# RESULTS

## Proof-of-concept

The first experiment intends to determine if SS-PSO is able to improve the performance of a standard S-PSO. For that purpose, three S-PSOs with different topologies have been implemented: $l$best with $k = 3$ (or $ring$) and two-dimensional square lattices with von Neumann ($k = 5$) and Moore neighborhood ($k = 9$). $G$best $k = n$ is not included in the comparisons because SS-PSO uses the neighborhood structure to decide how many and which particles to update: for instance, in the von Neumann topology ($k = 5$), five particles are updated. Since $g$best has $k = n$, the proposed strategy would update the entire population, that is, it would be equivalent to a S-PSO. Therefore, we have restricted the study to $l$best, von Neumann and Moore structures, labeling the algorithms, respectively, S-PSO$_{lbest}$, S-PSO$_{VN}$ and S-PSO$_{Moore}$.

Two sets of experiments were conducted. First, the algorithms were run for a specific amount of function evaluations (49,000 for $f_1$, $f_3$ and $f_6$, 980,000 for the remaining). After each run, the best solution was recorded. In the second set of experiments the algorithms were all run for 980,000 function evaluations or until reaching a function-specific stop criterion (given in Table 1). A success measure was defined as the number of runs in which an algorithm attains the stop criterion. This experimental setup is similar to those in *Kennedy & Mendes (2002)* and *Rada-Vilela, Zhang & Seah (2013)*. The dimension of

**Table 3 Median, minimum and maximum evaluations required to meet the criteria (50 runs).**

| | S-PSO$_{lbest}$ | | | S-PSO$_{VN}$ | | | S-PSO$_{Moore}$ | | |
|---|---|---|---|---|---|---|---|---|---|
| | Median | Min | Max | Median | Min | Max | Median | Min | Max |
| $f_1$ | 32,511.5 | 30,135 | 34,937 | 23,544.5 | 21,952 | 24,990 | **20,212** | 18,669 | 22,050 |
| $f_2$ | 365,270 | 313,551 | 403,858 | 217,854 | 188,111 | 242,893 | **173,117** | 142,688 | 194,530 |
| $f_3$ | 36,799 | 34,496 | 40,425 | 26,827 | 25,029 | 29,253 | **23,104** | 21,462 | 24,353 |
| $f_4$ | 77,518 | 21,462 | 866,173 | 15,582 | 9,604 | 74,872 | **13,524.0** | 7,448 | 49,392 |
| $f_5$ | 31,213 | 27,244 | 34,594 | 22,736 | 20,188 | 25,333 | **19,379.5** | 17,248 | 23,765 |
| $f_6$ | 18,865 | 5,243 | 145,334 | 12,323.5 | 3,626 | 80,213 | **7,105.0** | 3,822 | 39,788 |
| $f_7$ | 62,377 | 56,399 | 69,776 | 41,356 | 37,191 | 45,766 | **33,492** | 31,801 | 42,973 |
| $f_8$ | 35,206.5 | 31,556 | 39,249 | 24,206 | 22,834 | 28,928 | **20,923.0** | 19,012 | 24,794 |
| $f_9$ | – | – | – | 883,911 | 758,961 | 976,962 | **706,972** | 453,201 | 922,327 |
| $f_{10}$ | 33,001.5 | 30,331 | 37,926 | 24,157 | 21,805 | 26,460 | **21,021** | 18,865 | 29,939 |

Note:
Best median number of evaluations among the three algorithms shown in bold.

**Table 4 Success rates.**

| | S-PSO$_{lbest}$ | S-PSO$_{VN}$ | S-PSO$_{Moore}$ |
|---|---|---|---|
| $f_1$ | **50** | **50** | **50** |
| $f_2$ | **50** | **50** | **50** |
| $f_3$ | **50** | **50** | **50** |
| $f_4$ | 17 | **49** | **49** |
| $f_5$ | **50** | **50** | **50** |
| $f_6$ | **50** | **50** | **50** |
| $f_7$ | **50** | 47 | 34 |
| $f_8$ | **50** | **50** | **50** |
| $f_9$ | 6 | 9 | **47** |
| $f_{10}$ | **50** | **50** | 47 |

Note:
Best success rate among the three algorithms shown in bold.

the functions search space is $D = 30$ (except $f_6$, with $D = 2$). The results are in Table 2 (fitness), Table 3 (evaluations) and Table 4 (success rates). The best results among the three algorithms are shown in bold.

When compared to S-PSO$_{lbest}$, S-PSO$_{Moore}$ attains better solutions (considering median values of fitness distributions over 50 runs) in most of the functions and is faster (considering median values of evaluations required to meet the criteria) in every function. When compared to S-PSO$_{VN}$, S-PSO$_{Moore}$ is faster in every function and yields better median fitness values in unimodal functions.

In terms of success rates, S-PSO$_{Moore}$ clearly outperforms the other topologies in function $f_9$, and is much more efficient than S-PSO$_{lbest}$ in function $f_4$. These results are consistent with *Kennedy & Mendes (2002)*.

The algorithms were ranked by the Friedman test for each function. Table 5 shows the ranks according to the quality of solutions, while Table 6 shows the ranks according to

**Table 5 Fitness rank by Friedman test (with 0.05 significance level).** The table gives the rank of each algorithm and in parenthesis the algorithms to which the differences are significant according to the Friedman test.

|  | S-PSO$_{lbest}$ (1) | S-PSO$_{VN}$ (2) | S-PSO$_{Moore}$ (3) | P-value |
|---|---|---|---|---|
| $f_1$ | 3.0 (2) (3) | 2.0 (1) (3) | 1.0 (1) (2) | <0.0001 |
| $f_2$ | 3.0 (2) (3) | 2.0 (1) (3) | 1.0 (1) (2) | <0.0001 |
| $f_3$ | 3.0 (2) (3) | 2.0 (1) (3) | 1.0 (1) (2) | <0.0001 |
| $f_4$ | 2.98 (2) (3) | 1.47 (1) | 1.55 (1) | <0.0001 |
| $f_5$ | 1.57 (2) (3) | 2.03 (1) (2) | 2.40 (1) (3) | <0.0001 |
| $f_6$ | 2.24 (2) (3) | 1.94 (1) | 1.82 (1) | 0.00025 |
| $f_7$ | 1.57 (3) | 1.78 (3) | 2.65 (1) (3) | <0.0001 |
| $f_8$ | 2.44 (2) (3) | 1.96 (1) (3) | 1.60 (1) (2) | <0.0001 |
| $f_9$ | 2.96 (2) (3) | 1.98 (1) (3) | 1.06 (1) (2) | <0.0001 |
| $f_{10}$ | 1.61 (2) (3) | 1.99 (2) (3) | 2.40 (1) (2) | <0.0001 |

**Table 6 Convergence speed rank by Friedman test (with 0.05 significance level).** The table gives the rank of each algorithm and in parenthesis the algorithms to which the differences are significant according to the Friedman test.

|  | S-PSO$_{lbest}$ (1) | S-PSO$_{VN}$ (2) | S-PSO$_{Moore}$ (3) | P-value |
|---|---|---|---|---|
| $f_1$ | 3.0 (2) (3) | 1.99 (1) (3) | 1.01 (1) (2) | <0.0001 |
| $f_2$ | 3.0 (2) (3) | 1.98 (1) (3) | 1.02 (1) (2) | <0.0001 |
| $f_3$ | 3.0 (2) (3) | 1.99 (1) (3) | 1.01 (1) (2) | <0.0001 |
| $f_5$ | 3.0 (2) (3) | 1.96 (1) (3) | 1.04 (1) (2) | <0.0001 |
| $f_6$ | 2.35 (3) | 2.07 (3) | 1.58 (1) (2) | 0.00039 |
| $f_8$ | 3.00 (2) (3) | 2.00 (1) (3) | 1.00 (1) (2) | <0.0001 |

the convergence speed (only the functions on which the three algorithms attained the same success rates were considered in the ranking by convergence speed). Overall, S-PSO$_{Moore}$ ranks first in terms of solutions quality and convergence speed—see Fig. 1. Therefore, we conclude that the Moore structure is well suited for assessing the validity and relevance the SS-PSO.

Once the best network has been found for this particular set of problems, the next step was to compare synchronous and A-PSOs on the most efficient topology. For that purpose, we have implemented a SS-PSO$_{Moore}$ and tested it on the 10-function set under the same conditions described above. The results can be found in Table 7.

Table 8 gives a comparison between the performance of S-PSO$_{Moore}$ and SS-PSO$_{Moore}$ based on the numerical results and statistical analysis of those same results. The non-parametric Mann–Whitney test was used to compare the distribution of fitness values and number of evaluations to meet criteria of each algorithm in each function. The ranking of fitness distributions are significant at $P \leq 0.05$ for $f_1, f_2, f_3, f_6, f_7, f_9$, that is, in these functions, the null hypothesis that the two samples come from the same population is rejected. For the remaining functions ($f_5, f_8, f_{10}$), the null hypothesis is not rejected: the differences are not significant.

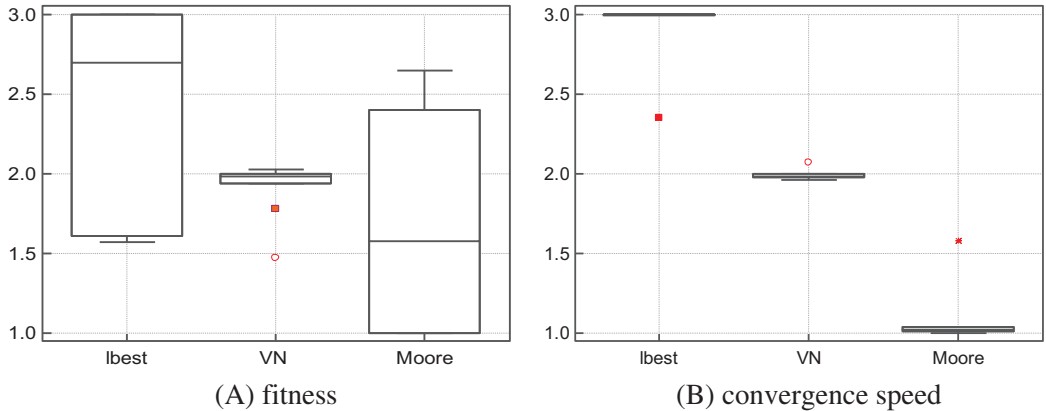

(A) fitness              (B) convergence speed

**Figure 1** S-PSO$_{lbest}$, S-PSO$_{VN}$ and S-PSO$_{Moore}$: solutions quality (A) and convergence speed (B) rank by the Friedman test.

**Table 7** SS-PSO$_{Moore}$ results: solutions quality, convergence speed and success rates.

|  | Fitness | | | Evaluations | | | |
|---|---|---|---|---|---|---|---|
|  | Median | Min | Max | Median | Min | Max | SR |
| $f_1$ | 5.42e−15 | 3.45e−16 | 6.49e−14 | 17,019 | 15,327 | 18,819 | 50 |
| $f_2$ | 7.18e−54 | 8.41e−60 | 4.87e−49 | 133,191 | 102,258 | 163,251 | 50 |
| $f_3$ | 2.99e−14 | 1.15e−15 | 2.97e−13 | 19,768.5 | 17,460 | 21,069 | 50 |
| $f_4$ | 5.12e+01 | 2.19e+01 | 1.04e+02 | 14,256 | 7,659 | 58,248 | 49 |
| $f_5$ | 7.40e−03 | 0.00e00 | 3.69e−02 | 16,884 | 14,814 | 24,291 | 50 |
| $f_6$ | 0.00e00 | 0.00e00 | 0.00e00 | 6,381 | 2,727 | 21,744 | 50 |
| $f_7$ | 0.00e00 | 0.00e00 | 1.32e−01 | 30,717 | 28,089 | 34,254 | 48 |
| $f_8$ | 8.88e−16 | 8.88e−16 | 1.33e−15 | 17,752.5 | 15,750 | 19,809 | 50 |
| $f_9$ | 1.01e−05 | 1.73e−08 | 7.11e−04 | 671,175 | 425,655 | 852,786 | 50 |
| $f_{10}$ | 3.70e−03 | 0.00e00 | 5.24e−01 | 17,662.5 | 15,669 | 27,252 | 48 |

**Table 8** Comparing S-PSO$_{Moore}$ and SS-PSO$_{Moore}$ with the Mann–Whitney test.

|  | $f_1$ | $f_2$ | $f_3$ | $f_4$ | $f_5$ | $f_6$ | $f_7$ | $f_8$ | $f_9$ | $f_{10}$ |
|---|---|---|---|---|---|---|---|---|---|---|
| Fitness | + | + | + | ≈ | ≈ | + | + | ≈ | + | ≈ |
| Evaluations | + | + | + | ≈ | + | ≈ | + | + | + | + |

Notes:
+If SS-PSO$_{Moore}$ ranks first in the Mann–Whitney test and the result is significant.
≈If the differences are not significant.

In terms of function evaluations, SS-PSO$_{Moore}$ is faster in the entire set of unimodal problems. In multimodal problems, SS-PSO$_{Moore}$ needs less evaluations in $f_5$, $f_6$, $f_7$ and $f_8$. Results of Mann–Whitney tests are significant at $P \leq 0.05$ for functions $f_1, f_2, f_3, f_5, f_7, f_8, f_9$ and $f_{10}$—see Table 8.

The success rates are similar, except for $f_7$ (in which SS-PSO clearly outperforms the standard algorithm) and $f_9$. In conclusion: empirical results, together with statistical tests, show that according to accuracy, speed and reliability, SS-PSO$_{Moore}$ outperforms

**Table 9 Results of SS-PSO variants: median, min, max and success rates (SR).**

| | SS-PSO$_{Moore}$ (*replace-best*) | | | | | | SR | SS-PSO$_{Moore}$ (*replace-random*) | | | | | | SR |
|---|---|---|---|---|---|---|---|---|---|---|---|---|---|---|
| | **Fitness** | | | **Evaluations** | | | | **Fitness** | | | **Evaluations** | | | |
| | Median | Min | Max | Median | Min | Max | | Median | Min | Max | Median | Min | Max | |
| $f_1$ | 4.09$e$−29 | 2.50$e$−33 | 2.00$e$+04 | 9,468 | 6,714 | 24,669 | 45 | 6.04$e$−14 | 7.86$e$−14 | 6.59$e$−12 | 18,972 | 16,425 | 20,781 | 50 |
| $f_2$ | 1.50$e$+04 | 4.12$e$−89 | 3.50$e$+04 | 66,307 | 64,251 | 68,364 | 2 | 8.33$e$−32 | 4.59$e$−34 | 5.00$e$+03 | 170,091 | 136,062 | 195,498 | 47 |
| $f_3$ | 3.01$e$−27 | 9.54$e$−34 | 1.00$e$+05 | 11,718 | 8,208 | 36,000 | 35 | 1.66$e$−12 | 1.30$e$−13 | 2.25$e$−11 | 21,118 | 19,548 | 23,283 | 50 |
| $f_4$ | 1.30$e$+02 | 7.46$e$+01 | 2.00$e$+02 | 15,192 | 8,964 | 108,495 | 9 | 5.62$e$+01 | 2.39$e$+01 | 8.76$e$+01 | 11,052 | 5,679 | 23,571 | 50 |
| $f_5$ | 3.08$e$−02 | 0.00$e$00 | 1.81$e$+02 | 10,287 | 8,694 | 26,838 | 12 | 0.00$e$00 | 0.00$e$00 | 8.33$e$−02 | 19,849.5 | 17,748 | 26,739 | 36 |
| $f_6$ | 3.59$e$−04 | 0.00$e$00 | 9.72$e$−03 | 39,811.5 | 1,242 | 140,247 | 38 | 0.00$e$00 | 0.00$e$00 | 9.72$e$−03 | 8,460 | 3,276 | 62,091 | 50 |
| $f_7$ | 7.52$e$00 | 2.64$e$00 | 1.57$e$+01 | – | – | – | 0 | 1.58$e$−03 | 0.00$e$00 | 2.48$e$00 | 33,912 | 31,239 | 41,211 | 30 |
| $f_8$ | 2.28$e$00 | 8.86$e$−16 | 3.84$e$00 | 20,898 | 13,158 | 28,764 | 6 | 1.11$e$−15 | 8,86$e$−16 | 1.33$e$−15 | 19,822.5 | 18,252 | 25,416 | 50 |
| $f_9$ | 1.06$e$−01 | 1.98$e$−03 | 1.53$e$+04 | 902,407 | 812,736 | 949,590 | 12 | 1.64$e$−04 | 1.44$e$−06 | 6.01$e$+01 | 736,713 | 546,858 | 891,432 | 49 |
| $f_{10}$ | 1.04$e$+01 | 0.00$e$00 | 4.04$e$+02 | 16,065 | 8,388 | 23,742 | 2 | 3.70$e$−03 | 0.00$e$00 | 5.09$e$−01 | 21,915 | 18,567 | 50,607 | 39 |

S-PSO$_{Moore}$ in most of the benchmark functions selected for this test, while not being outperformed in any case.

## Update strategy

The preceding tests show that the steady state update strategy when implemented in a PSO structured in a lattice with Moore neighborhood improves its performance. The following experiment aims at answering an important question: what is the major factor in the performance enhancement? Is it the steady state update, or instead the particles that are updated?

In order to investigate this issue, two variants of SS-PSO were implemented: one that updates the best particle and its neighbors (*replace-best*); and another that updates a randomly selected particle and its neighbors (*replace-random*). The algorithms were tested on the same set of benchmark functions and compared the proposed SS-PSO$_{Moore}$ (or *replace-worst*). Results are in Table 9.

*Replace-best* update strategy is outperformed by *replace-worst* SS-PSO. With the exception of $f_1$ and $f_3$, the quality of solutions is degraded when compared to the proposed SS-PSO. However, success rates are considerably lower in most functions, including $f_1$ and $f_3$. Please note that functions $f_1$ and $f_3$ are unimodal and therefore they can be easily solved by hill-climbing and greedy algorithms. It is not surprising that a greedy selective strategy like SS-PSO with *replace-best* can find very good solutions in some runs. However, for more difficult problem, *replace-best* is clearly unable to find good solutions.

As for *replace-random*, it improves S-PSO in some functions, but in general is not better than *replace-worst*: replace-random SS-PSO is less accurate and slower in most of the functions. The Friedman test shows that SS-PSO with *replace-worst* strategy ranks first in terms of solutions quality—see Fig. 2.

Table 10 compares *replace-random* and *replace-worst* with the assistance of Mann–Whitney statistical tests. Except for $f_4$, *replace-worst* is significantly more efficient

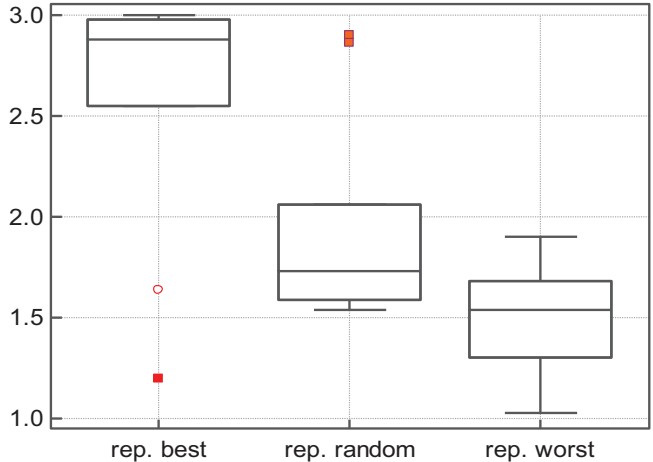

**Figure 2 Fitness rank by Friedman test.**

**Table 10 Comparing *replace-worst* and *replace-random* with the Mann-Whitney test.**

|  | $f_1$ | $f_2$ | $f_3$ | $f_4$ | $f_5$ | $f_6$ | $f_7$ | $f_8$ | $f_9$ | $f_{10}$ |
|---|---|---|---|---|---|---|---|---|---|---|
| Fitness | + | + | + | ≈ | ≈ | ≈ | + | + | + | ≈ |
| Evaluations | + | + | + | − | + | + | + | + | + | + |

**Notes:**
+If *replace-worst* ranks first in the Mann–Whitney test and the result is significant.
−If *replace-random* ranks first and the result is significant.
≈If the differences are not significant.

than *replace-random*. The experiment demonstrates that selective pressure imposed on the least fit individuals is the major factor in the performance of SS-PSO.

## Scalability

The proof-of-concept showed that SS-PSO outperforms S-PSO in most of the functions in the test set, and the previous experiment demonstrates that the major factor in the performance enhancement is the pressure on the least fit particles. However, only instances of the problems with $D = 30$ have been tested; therefore, another question arises at this point: does the improvement shown by SS-PSO hold for a wide range of problem sizes? In order to answer that question, we have conducted a scalability study: the algorithms were tested on the same set functions but with $D$ ranging from 10 to 50 (except $f_6$, which is a two-dimensional function and for that reason was excluded from this test).

As in previous experiments, the algorithms were first run for a limited amount of function evaluations and the best fitness values were recorded. Then, the algorithms were all run for 980,000 evaluations or until reaching a function-specific stop criterion. The number of iterations required to meet the criterion was recorded and statistical measures were taken over 50 runs. (Function $f_{10}$ has not been tested for dimensions 20 and 40 because the CEC2005 benchmark, from where the orthogonal rotational matrices $M$ have been taken, does not provide the matrices for those dimensions).

Table 11 shows the median best fitness values attained by each algorithm on each instance of the problems and Table 12 shows the success rates. In terms of quality of

**Table 11 Solutions quality with different problem dimension.**

|  | D = 10 | | D = 20 | | D = 30 | | D = 40 | | D = 50 | |
|---|---|---|---|---|---|---|---|---|---|---|
|  | S-PSO | SS-PSO | S-PSO | SS-PSO | S-PSO | SS-PSO | S-PSO | SS-PSO | S-PSO | SS-PSO |
| $f_1$ | 1.06e−37 | **2.71e−47** | 1.87e−19 | **5.72e−24** | 1.04e−11 | **7.83e−15** | 7.15e−08 | **2.96e−10** | 3.69e−05 | **2.01e−10** |
| $f_2$ | **0.00e00** | **0.00e00** | 4.63e−82 | **1.37e−89** | 1.17e−30 | **9.52e−54** | 1.18e−13 | **1.10e−20** | **1.36e−06** | 2.36e−06 |
| $f_3$ | 1.57e−40 | **0.00e00** | 2.08e−19 | **3.37e−24** | 2.76e−11 | **1.58e−14** | 8.77e−07 | **2.58e−09** | 4.59e−04 | **3.19e−06** |
| $f_4$ | 1.99e00 | 1.99e00 | 2.09e+01 | **2.04e+01** | 6.17e+01 | **5.12e+01** | **1.01e+02** | 1.06e+02 | 1.70e+02 | **1.37e+02** |
| $f_5$ | **2.83e−02** | 3.60e−02 | **8.63e−03** | 1.11e−02 | **0.00e00** | 7.40e−03 | **0.00e00** | 7.40e−03 | **0.00e00** | **0.00e00** |
| $f_7$ | **0.00e00** | **0.00e00** | **0.00e00** | **0.00e00** | 9.03e−04 | **0.00e00** | 9.03e−04 | **3.39e−04** | 1.34e−01 | **2.15e−02** |
| $f_8$ | 4.44e−16 | 4.44e−16 | 8.88e−16 | 8.88e−16 | 8.88e−16 | 8.88e−16 | 1.33e−15 | 1.33e−15 | 1.33e−15 | 1.33e−15 |
| $f_9$ | **0.00e00** | **0.00e00** | 1.92e−10 | **1.32e−10** | 9.8e−05 | **1.01e−05** | 6.18e+01 | **3.40e+01** | 1.34e+03 | **1.70e+03** |
| $f_{10}$ | 3.20e−02 | 3.20e−02 | – | – | 7.40e−03 | 7.40e−03 | – | – | 0.00e00 | 0.00e00 |

**Note:**
Best median fitness among the two algorithms shown in bold.

**Table 12 Success rates with different problem dimension.**

|  | D = 10 | | D = 20 | | D = 30 | | D = 40 | | D = 50 | |
|---|---|---|---|---|---|---|---|---|---|---|
|  | S-PSO | SS-PSO | S-PSO | SS-PSO | S-PSO | SS-PSO | S-PSO | SS-PSO | S-PSO | SS-PSO |
| $f_1$ | 50 | 50 | 50 | 50 | 50 | 50 | 50 | 50 | 50 | 50 |
| $f_2$ | 50 | 50 | 50 | 50 | 50 | 50 | 43 | 50 | 32 | 48 |
| $f_3$ | 50 | 50 | 50 | 50 | 50 | 50 | 50 | 50 | 50 | 50 |
| $f_4$ | 50 | 50 | 50 | 50 | 49 | 49 | 25 | 21 | 0 | 3 |
| $f_5$ | 40 | 37 | 49 | 50 | 50 | 47 | 50 | 49 | 50 | 50 |
| $f_7$ | 50 | 50 | 49 | 49 | 34 | 48 | 8 | 35 | 4 | 19 |
| $f_8$ | 50 | 50 | 50 | 50 | 50 | 50 | 50 | 50 | 46 | 50 |
| $f_9$ | 50 | 50 | 50 | 50 | 47 | 50 | 0 | 0 | 0 | 0 |
| $f_{10}$ | 44 | 35 | – | – | 46 | 48 | – | – | 48 | 49 |

**Note:**
Best success rates among the two algorithms shown in bold.

solutions, the performance patterns observed with $D = 30$ are maintained: the strategy does not introduce scalability difficulties. As for the success rates, except for a few instances, SS-PSO attains better or equal success rates.

The convergence speed has been graphically represented for better assessment of the effects of growing problem size—see Fig. 3. The graphs show that the proposed strategy does not introduce scalability difficulties (other than the ones intrinsic to standard PSOs). It also shows that, in general, SS-PSO is faster than S-PSO.

## Parameter sensitivity

Particle swarm optimization performance can be severely affected by the parameter values. The inertia weight and acceleration coefficients must be tuned in order to balance exploration and exploitation: if far from the optimal values, convergence speed and/or solution quality can be significantly reduced. Population size also influences the performance of population-based metaheuristics: larger populations help to maintain

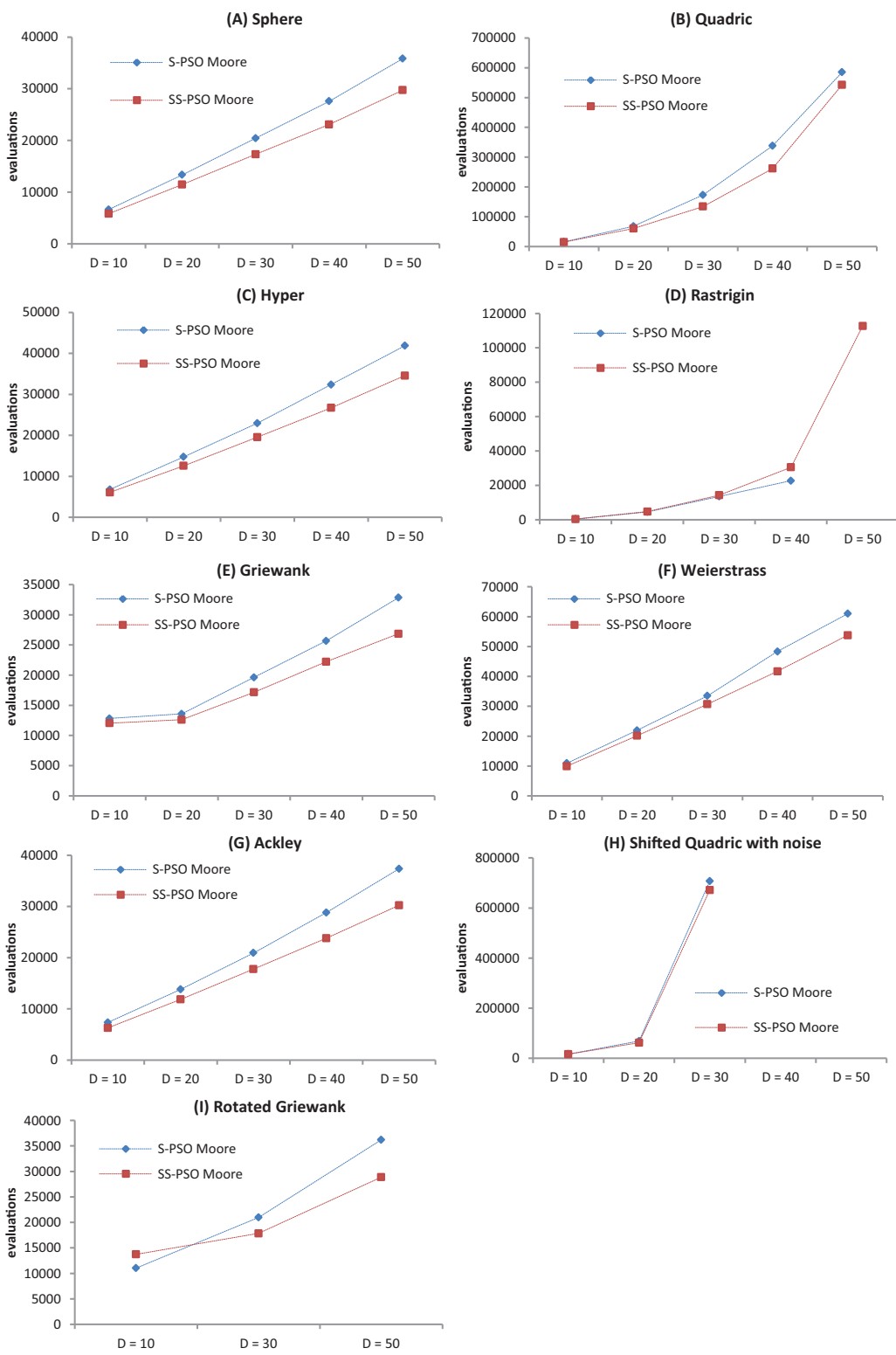

**Figure 3 Convergence speed versus problem dimension for Sphere (A), Quadric (B), Hyper (C), Rastrigin (D), Griewank (E), Weierstrass (F), Ackley (G), Shifted Quadric with Noise (H) and Rotated Griewank (I) benchmark functions.**

diversity, but they slow down convergence speed; on the other hand, smaller populations are faster but they are more likely to converge to local optima.

Furthermore, PSOs empirical studies usually depend on a single set of parameters for several functions with different characteristics. This is the case of this paper, in which a typical parameter setting has been used for evaluating the performance of the PSOs. That set of parameters is not expected to be the optimal tuning for every function, but instead a compromised solution to avoid the exponential growth of experimental procedures.

For these reasons, when testing a new PSO, it is important to investigate its sensitivity to the parameter values. With that purpose in mind, the following experimental procedure has been designed.

Synchronous PSO and SS-PSO were tested on function $f_1$ (unimodal), $f_2$ (multimodal), $f_9$ (shifted and noisy) and $f_{10}$ (rotated) with the following parameter values: inertia weight was set to 0.6798, 0.7048, 0.7298, 0.7548 and 0.7798, while acceleration coefficients and population size remained fixed at 1.494 and 49; then, $c_1$ and $c_2$ were set to 1.294, 1.394, 1.494, 1.594 and 1.694 while $\omega$ and $\mu$ remained fixed at 0.7298 and 49, respectively; finally, population size was set to 36, 49 and 64, while $\omega$ and the acceleration coefficients were set to 0.7298 and 1.4962. The results are depicted in Figs. 4–7.

The graphics show that the performance indeed varies with the parameter values, as expected. In the case of function $f_1$, other parameter settings attain better results than the ones used in previous section. However, the relative performance of S-PSO and SS-PSO maintains throughout the parameters ranges. In functions $f_8$, $f_9$ and $f_{10}$, the quality of solutions is in general maximized by $\omega$ and $c$ values around the ones used in previous sections. Convergence speed, in general, improves with lower $\omega$, $c$ and $\mu$ values.

As seen in Fig. 1, S-PSO$_{\text{Moore}}$ ranks first in terms of solutions quality and convergence speed when compared to ring and von Neumann topologies. Although not a parameter in the strict sense of the term, the network topology is a design choice that significantly affects the performance of the algorithm: *Kennedy & Mendes (2002)* investigated several types of networks and recommend the use of von Neumann lattices; *Fernandes et al. (2018)* tested regular graphs and concluded that convergence speed improves with the degree of connectivity but success rates are in general degraded when $k$ is above nine (equivalent to Moore neighborhood) and a that good compromise is achieved with $5 \leq k \leq 13$.

In order to study the performance of SS-PSO with different network topologies, regular graphs have been constructed with the following procedure: starting from a ring structure with $k = 3$, the degree is increased by linking each individual to its neighbors' neighbors, creating a set of regular graphs with $k = \{3, 5, 7, 9, 11 \ldots, \mu\}$, as exemplified in Fig. 8 for population size 7. Parameters $c_1$ and $c_2$ were set to 1.494 and $\omega$ to 0.7298 and population size $\mu$ was set to 33. The algorithms were all run for 660,000 function evaluations or until reaching the function-specific stop criterion given in Table 1. Each algorithm has been executed 50 times with each function and statistical measures were taken over those 50 runs.

Figure 9 shows the success rates and convergence speed of SS-PSO structured by topologies with varying $k$. Convergence generally improves with $k$, achieving optimal values for $13 \leq k \leq 25$ in most of the functions. However, as seen in Fig. 9A, best success rates are achieved when $7 \leq k \leq 13$ (except $f_{10}$, for which $k = 5$ is the best topology).

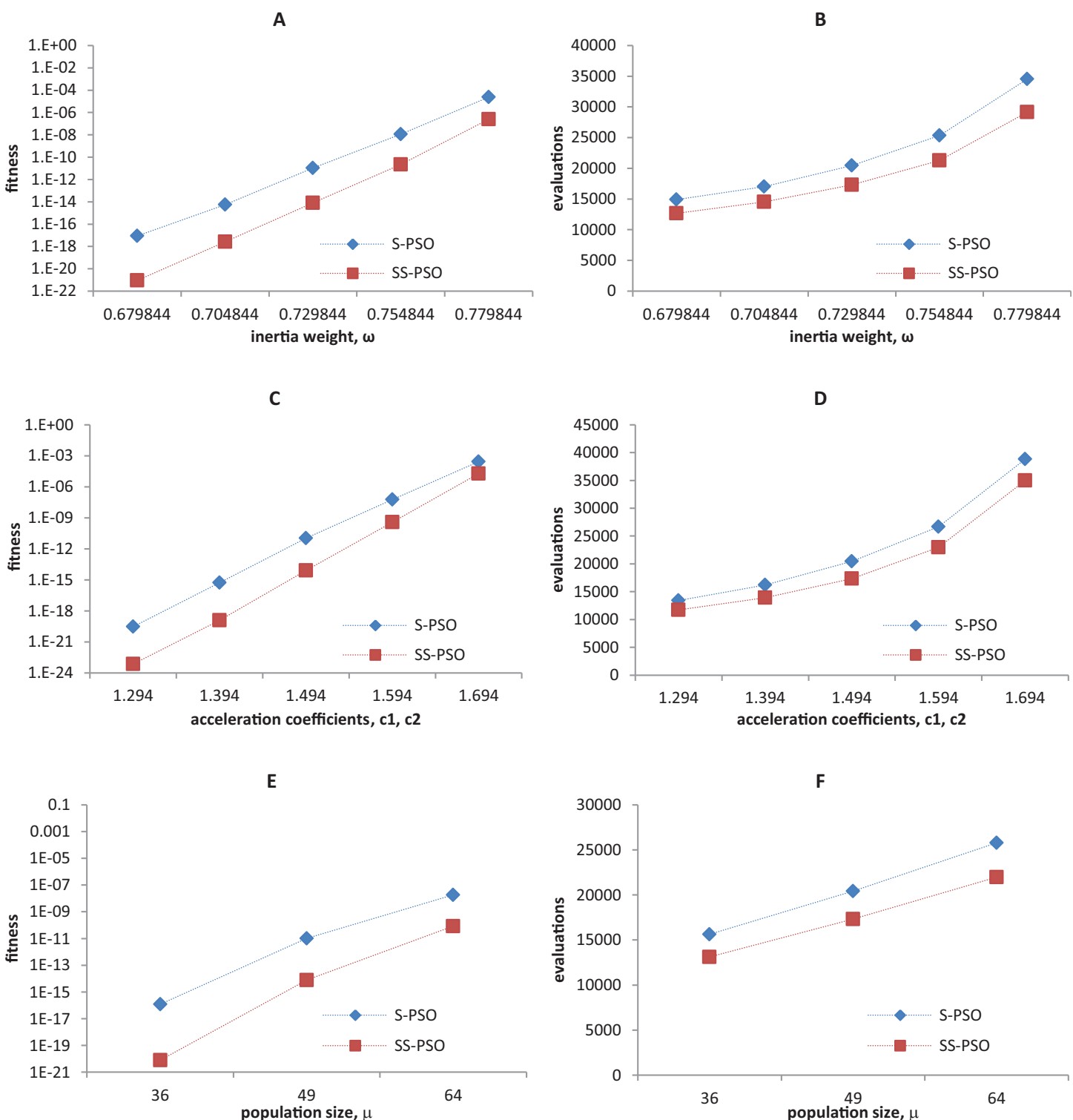

**Figure 4** Fitness (A, C, E) and number of evaluations sensitivity (B, D, F) on sphere function ($f_1$) to inertia weight (A–B), acceleration coefficients (C–D) and population size (E–F).

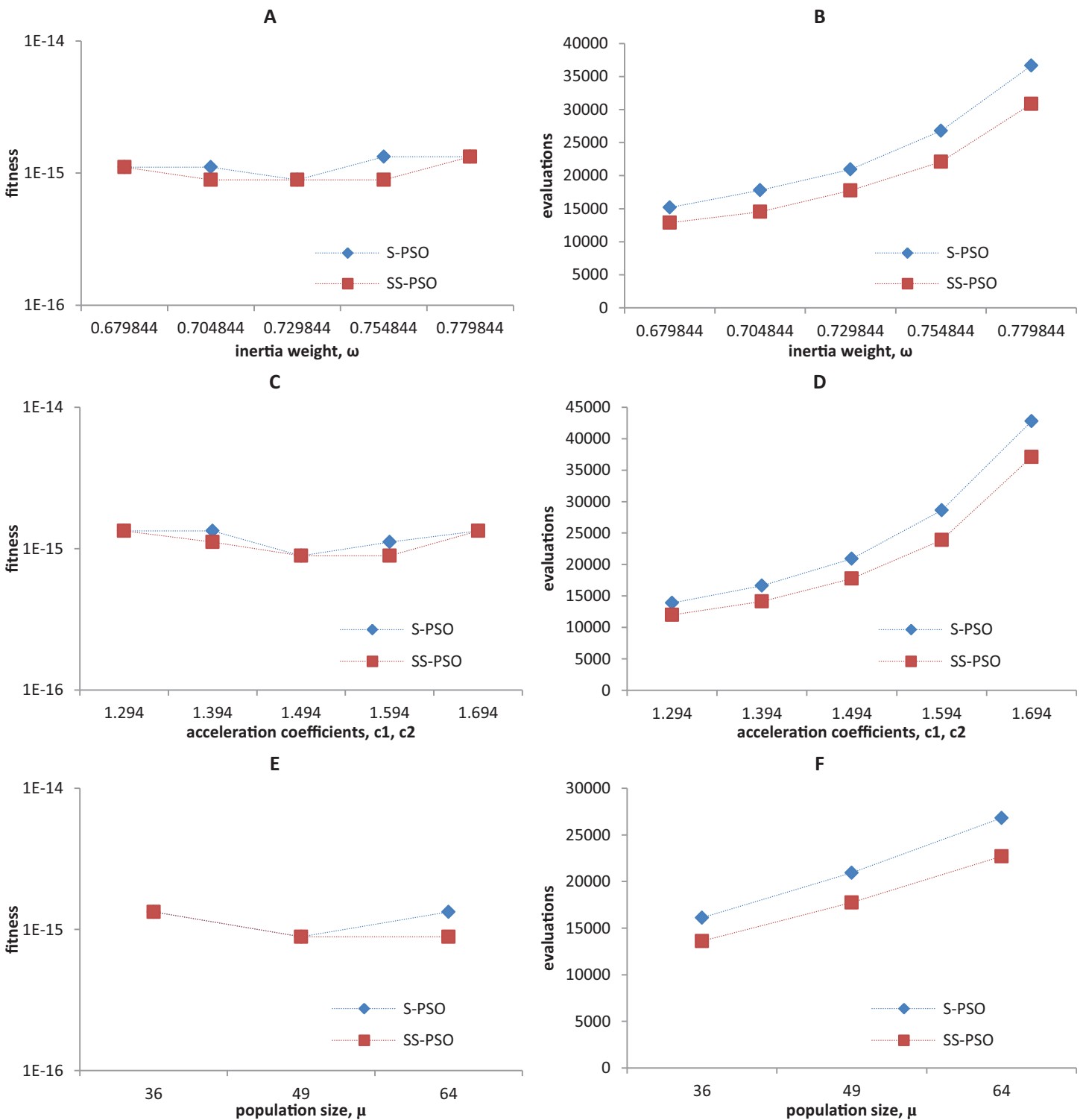

**Figure 5** Fitness (A, C, E) and number of evaluations sensitivity (B, D, F) on Ackley function ($f_8$) to inertia weight (A–B), acceleration coefficients (C–D) and population size (E–F).

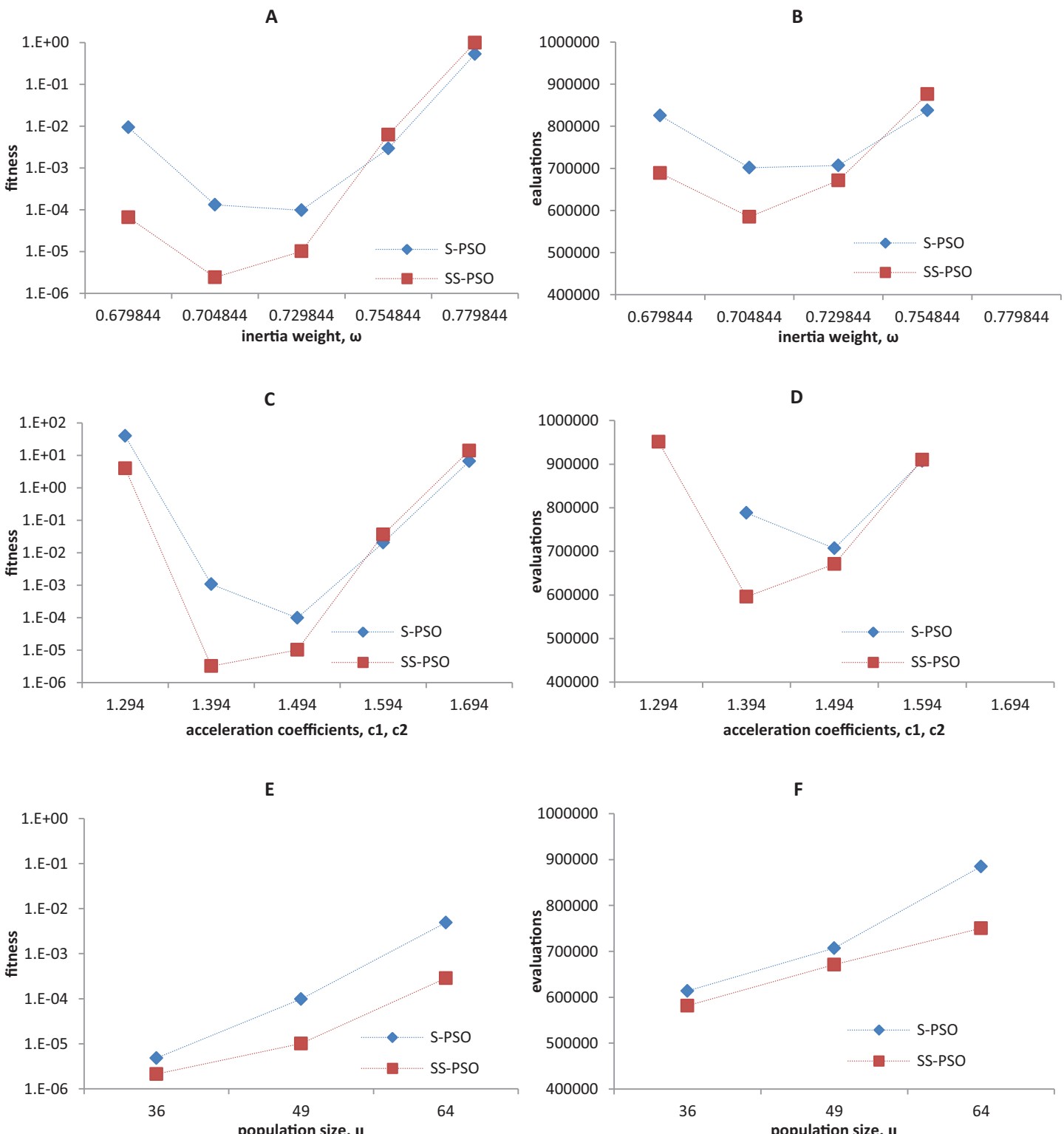

**Figure 6** Fitness (A, C, E) and number of evaluations sensitivity (B, D, F) on shifted quadric with noise function ($f_9$) to inertia weight (A–B), acceleration coefficients (C–D) and population size (E–F).

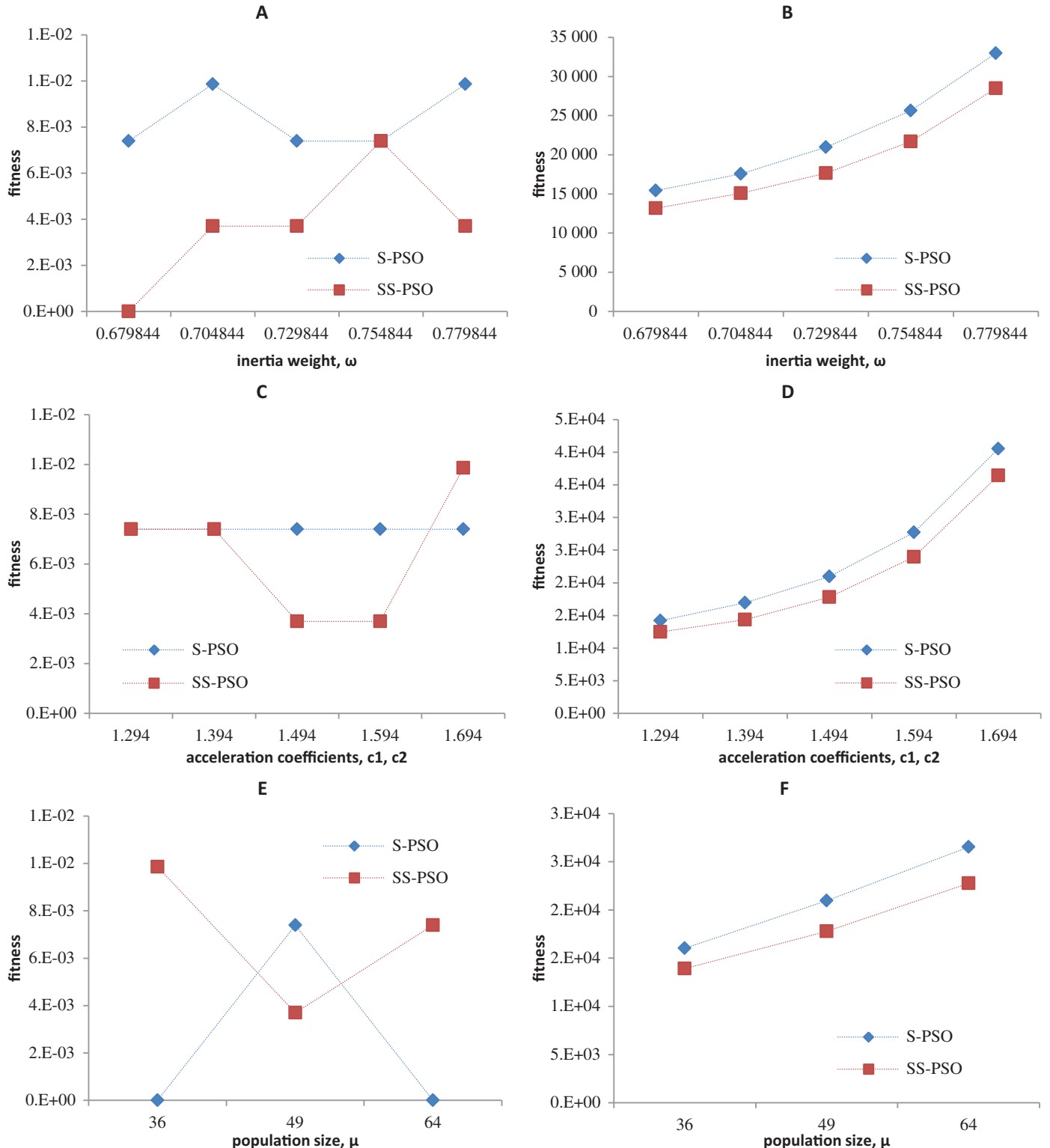

**Figure 7** Fitness (A, C, E) and number of evaluations sensitivity (B, D, F) on Griewank function ($f_{10}$) to inertia weight (A–B), acceleration coefficients (C–D) and population size (E–F).

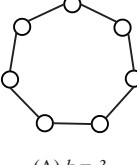
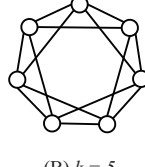
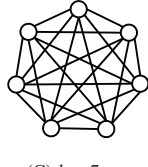

(A) $k = 3$       (B) $k = 5$       (C) $k = 7 = \mu$

**Figure 8 Regular graphs with population size $\mu = 7$ and $k = 3$ (A), $k = 5$ (B) and $k = 7 = \mu$ (C).**

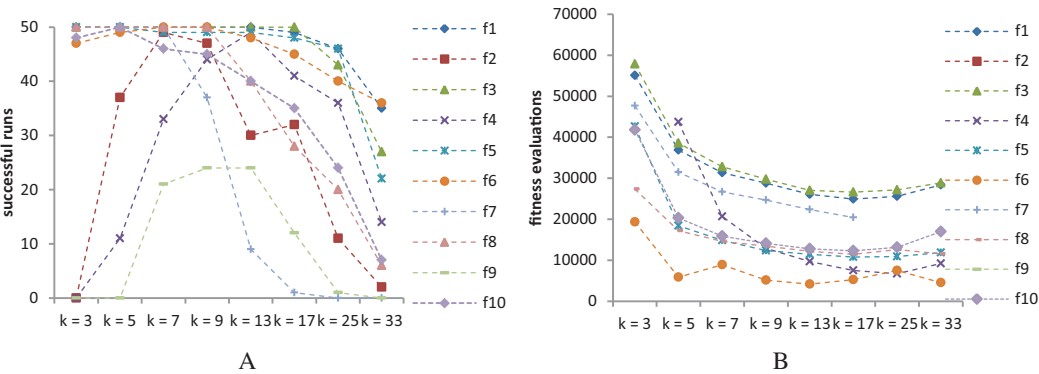

**Figure 9 SS-PSO with different topologies.** (A) Success rates. (B) Mean fitness evaluations to a solution.

These conclusions are similar to those by *Fernandes et al. (2018)* related to the standard PSO and are coincident with the typical rule of thumb for PSOs: highly connected topologies are faster but less reliable, while topologies with lower connectivity require more evaluations to meet the convergence criteria but converge more often to the solution.

Please remember that we are not trying to find the best set of parameters for each function. The most important conclusions here is that SS-PSO does not seem to be more sensitive to the parameters than S-PSO, displaying similar patterns when varying $\omega$, $c_1$ and $c_2$ and $\mu$, and that the performance enhancement brought by SS-PSO is observed on a reasonably wide range of parameter values.

### Time-varying parameters

An alternative approach to parameter tuning is to let the parameters values change during the run, according to deterministic or adaptive rules. In order to avoid tuning effort and adapt the balance between local and global search to the search stage, *Shi & Eberhart (1999)* proposed a linearly time-varying inertia weight: starting with an initial and pre-defined value, the parameter value decreases linearly with time, until it reaches the minimum value. The variation rule is given by Eq. (3):

$$\omega(t) = (\omega_1 - \omega_2) \times \frac{(\max\_t - t)}{\max\_t} + \omega_2 \tag{3}$$

where $t$ is the current iteration, $\max\_t$ is the maximum number of iterations, $\omega_1$ the inertia weigh initial value and $\omega_2$ its final value.

**Table 13  PSO-TVAC results.**

| | Fitness | | | Evaluations | | | SR |
|---|---|---|---|---|---|---|---|
| | Median | Min | Max | Median | Min | Max | |
| $f_1$ | 2.85e−21 | 2.55e−22 | 1.84e−20 | 11,956 | 11,221 | 13,181 | 50 |
| $f_2$ | 4.47e−51 | 1.23e−54 | 5.00e03 | 208,740 | 185,514 | 238,532 | 49 |
| $f_3$ | 3.87e−21 | 3.01e−22 | 1.57e−19 | 13,769 | 12,740 | 16,121 | 50 |
| $f_4$ | 3.08e+01 | 1.11e+01 | 5.8e+01 | **31,114** | 16,661 | 59,388 | 50 |
| $f_5$ | **0.00e00** | 0.00e00 | 4.91e−02 | 15,141 | 12,642 | 91,238 | 50 |
| $f_6$ | **0.00e00** | 0.00e00 | 0.00e00 | 11,956 | 5,145 | 38,612 | 49 |
| $f_7$ | **0.00e00** | 0.00e00 | 1.64e−01 | 35,280 | 31,017 | 42,336 | 49 |
| $f_8$ | **7.55e−15** | 4.00e−15 | 7.55e−15 | 21,070 | 17,346 | 29,988 | 50 |
| $f_9$ | 6.14e−09 | 1.74e−09 | 6.28e−06 | **227,066** | 199,528 | 287,042 | 47 |
| $f_{10}$ | 7.40e−03 | 0.00e00 | 5.24e−01 | 18,620 | 14,602 | 87,220 | 42 |

Note:
Medians are shown in bold if PSO-TVAC provides similar or better results than SS-PSO-TVAC (Table 14).

Later, *Ratnaweera, Halgamuge & Watson (2004)* tried to improve Shi and Eberhart's PSO with time-varying inertia weight (PSO-TVIW) using a similar concept applied to the acceleration coefficients. In the PSO with time-varying acceleration coefficients PSO (PSO-TVAC) the parameters $c_1$ and $c_2$ change during the run according to the following equations:

$$c_1 = \left(c_{1f} - c_{1i}\right) \times \frac{t}{\max\_t} + c_{1i} \tag{4}$$

$$c_2 = \left(c_{2f} - c_{2i}\right) \times \frac{t}{\max\_t} + c_{2i} \tag{5}$$

where $c_{1i}$, $c_{1f}$, $c_{2i}$, $c_{2f}$ are the acceleration coefficients initial and final values.

The experiments in this section compare PSO-TVAC with SS-PSO-TVAC (i.e., PSO-TVAC with the steady-state update strategy). Parameters $\omega_1$ and $\omega_2$ were set to 0.75 and 0.5. The acceleration coefficient $c_1$ initial and final values were set to 2.5 and 0.5 and $c_2$ ranges from 0.5 to 2.5, as suggested by *Ratnaweera, Halgamuge & Watson (2004)*. The results are in Table 13 (PSO-TVAC) and Table 14 (SS-PSO-TVAC).

Table 15 compares the algorithms using Mann–Whitney tests. SS-PSO-TVAC improves PSO-TVAC in every unimodal function in terms of accuracy and convergence speed and it is significantly faster in functions $f_6$, $f_7$, $f_8$ and $f_{10}$ while attaining similar results. PSO-TVAC only outperforms SS-PSO-TVAC in the noisy $f_9$ function. These results show that the steady state version of PSO-TVAC is able to improve the convergence speed of the original algorithm in several types of fitness landscapes. Furthermore, SS-PSO-TVAC achieves more accurate solutions in the unimodal problems.

## Comprehensive learning PSO

The following experiment aims at comparing the proposed SS-PSO with the CLPSO (*Liang et al., 2006*; *Lynn & Suganthan, 2015*). CLPSO uses an alternative velocity updating equation:

$$v_{i,d}(t) = \omega \times v_{i,d}(t-1) + c \times r \times \left(p_{fi(d),d} - x_{i,d}(t-1)\right) \tag{6}$$

**Table 14  SS-PSO-TVAC results.**

| | Fitness | | | Evaluations | | | SR |
|---|---|---|---|---|---|---|---|
| | Median | Min | Max | Median | Min | Max | |
| $f_1$ | **7.85e−26** | 4.82e−27 | 2.35e−24 | **10,417** | 9,126 | 11,322 | 50 |
| $f_2$ | **5.18e−63** | 2.30e−67 | 5.77e−60 | **190,458** | 168,282 | 226,062 | 50 |
| $f_3$ | **1.66e−25** | 7.76e−27 | 9.14e−24 | **11,925** | 10,422 | 13,923 | 50 |
| $f_4$ | **3.48e+01** | 1.89e01 | 7.46e+01 | 38,043 | 22,032 | 108,927 | 50 |
| $f_5$ | **0.00e00** | 0.00e00 | 4.42e−02 | **13,662** | 9,963 | 56,421 | 49 |
| $f_6$ | **0.00e00** | 0.00e00 | 0.00e00 | **8,421** | 2,547 | 26,325 | 49 |
| $f_7$ | **0.00e00** | 0.00e00 | 2.62e−01 | **31,752** | 28,323 | 41,193 | 43 |
| $f_8$ | **7.55e−15** | 4.00e−15 | 7.55e−15 | **18,756** | 14,958 | 23,904 | 49 |
| $f_9$ | **5.41e−09** | 6.37e−10 | 5.80e−03 | 315,792 | 192,906 | 476,532 | 48 |
| $f_{10}$ | **0.00e00** | 0.00e00 | 3.93e−02 | **15,948** | 12,762 | 75,510 | 40 |

**Note:**
 Medians are shown in bold if SS-PSO-TVAC provides similar or better results than PSO-TVAC (Table 13).

**Table 15  Comparing SS-PSO-TVAC and PSO-TVAC with the Mann-Whitney test.**

| | $f_1$ | $f_2$ | $f_3$ | $f_4$ | $f_5$ | $f_6$ | $f_7$ | $f_8$ | $f_9$ | $f_{10}$ |
|---|---|---|---|---|---|---|---|---|---|---|
| Fitness | + | + | + | ≈ | ≈ | ≈ | ≈ | ≈ | ≈ | ≈ |
| Evaluations | + | + | + | ≈ | ≈ | + | + | + | − | + |

**Notes:**
 +If SS-PSO-TVAC ranks first in the Mann–Whitney test and the result is significant.
 −If PSO-TVAC ranks first and the results is significant.
 ≈If the differences are not significant.

where $\overrightarrow{f_i} = (f_i(1), f_i(2), \ldots f_i(D))$ defines which particle's best solutions particle $i$ should follow. Hence, the term $p_{fi(d),d}$ can refer to the corresponding dimension of any particle's best found solution so far. The decision depends on a probability $p_c$, different for each particle and computed a priori. Following the guidelines and parameters in *Liang et al. (2006)*, CLPSO and SS-CLPSO have been implemented and tested in the set of 10 benchmark functions.

Comprehensive-learning PSO performance is strongly dependent on the refreshing gap parameter $m$, which defines the number of generations during which the particles are allowed to learn from $f_i$ without improving their fitness. After $m$ generations without fitness improvement, $f_i$ is reassigned. In order to make fair comparisons, parameter $m$ was first optimized for each function. The other parameters were set as in *Liang et al. (2006)*. Then, SS-CLPSO was tuned using the same parameter setting as the corresponding CLPSO.

The results are in Tables 16 and 17 and statistical analysis is in Table 18. On the one hand, the results show that, in general, a steady-state strategy applied to CLPSO does not improve the performance of the algorithm. On the other hand, SS-CLPSO does not degrade the general behavior of CLPSO. Please note that CLPSO does not use a traditional topology. In this case, to construct SS-CLPSO, we use a Moore neighborhood to decide which particles to update along with the least fit individuals, but, unlike SS-PSO or SS-PSO-TVAC, the structure does not define the information flow within the swarm.

**Table 16  CLPSO results.**

| | Fitness | | | Evaluations | | | |
|---|---|---|---|---|---|---|---|
| | Median | Min | Max | Median | Min | Max | SR |
| $f_1$ | **9.59e−07** | 4.00e−07 | 2.23e−06 | **34,848** | 33,355 | 35,909 | 50 |
| $f_2$ | **2.16e−01** | 8.66e−02 | 5.46e−01 | – | – | – | – |
| $f_3$ | 2.31e−06 | 1.18e−06 | 4.96e−06 | 36,777 | 35,665 | 37,972 | 50 |
| $f_4$ | 4.97e00 | 1.99e00 | 1.20e+01 | **115,701** | 94,674 | 129,493 | 50 |
| $f_5$ | **0.00e00** | 0.00e00 | 9.74e−13 | 199,537 | 164,774 | 243,806 | 50 |
| $f_6$ | **6.49e−06** | 7.69e−09 | 1.13e−04 | **81,149** | 37,710 | 96,320 | 31 |
| $f_7$ | **8.67e−13** | 3.48e−13 | 1.61e−12 | 430,069 | 418,700 | 440,035 | 50 |
| $f_8$ | **7.55e−15** | 4.00e−15 | 7.55e−15 | **282,613** | 275,897 | 285,290 | 50 |
| $f_9$ | **4.36e−01** | 1.51e−01 | 1.11e00 | – | – | – | – |
| $f_{10}$ | **0.00e00** | 0.00e00 | 2.26e−14 | 173,346 | 151,269 | 229,975 | 50 |

Note:
    Medians are shown in bold if CLPSO provides similar or better results than SS-CLPSO (Table 17).

**Table 17  SS-CLPSO results.**

| | Fitness | | | Evaluations | | | |
|---|---|---|---|---|---|---|---|
| | Median | Min | Max | Median | Min | Max | SR |
| $f_1$ | 1.33e−06 | 2.99e−07 | 4.98e−06 | 35,998 | 34,063 | 37,956 | 50 |
| $f_2$ | 5.14e−01 | 1.71e−01 | 1.44e00 | – | – | – | – |
| $f_3$ | **1.46e−06** | 4.82e−07 | 7.44e−06 | **36,079** | 33,177 | 37,961 | 50 |
| $f_4$ | **4.09e+00** | 1.04e00 | 1.05e+01 | 190,310 | 147,544 | 217,855 | 50 |
| $f_5$ | **0.00e00** | 0.00e00 | 2.16e−14 | **181,779** | 137,821 | 225,172 | 50 |
| $f_6$ | 6.64e−06 | 3.10e−07 | 6.90e−05 | 86,058 | 45,530 | 97,936 | 28 |
| $f_7$ | 6.26e−11 | 3.27e−12 | 1.69e−08 | **409,351** | 393,553 | 423,387 | 50 |
| $f_8$ | **7.55e−15** | 4.00e−15 | 7.55e−15 | 358,407 | 344,448 | 374,581 | 50 |
| $f_9$ | 7.70e−01 | 3.26e−01 | 6.16e01 | – | – | – | – |
| $f_{10}$ | **0.00e00** | 0.00e00 | 1.04e−13 | **152,818** | 122,165 | 207,094 | 50 |

Note:
    Medians are shown in bold if SS-CLPSO provides similar or better results than CLPSO (Table 16).

**Table 18  Comparing SS-PSO and CLPSO with the Mann-Whitney test.**

| | $f_1$ | $f_2$ | $f_3$ | $f_4$ | $f_5$ | $f_6$ | $f_7$ | $f_8$ | $f_9$ | $f_{10}$ |
|---|---|---|---|---|---|---|---|---|---|---|
| Fitness | ≈ | ≈ | ≈ | ≈ | ≈ | ≈ | − | ≈ | ≈ | ≈ |
| Eval. | ≈ | − | ≈ | − | + | ≈ | + | − | − | + |

Notes:
    +If SS-PSO ranks first in the Mann–Whitney test and the result is significant.
    −If CLPSO ranks first and the results is significant.
    ≈If the differences are not significant.

Since neighboring particles communicate and use each other's information, they tend to travel through similar regions of the landscape, but in CLPOS there is not necessarily a relationship between the particles in the set and this clustering behavior is not present. For a steady-state strategy to take full advantage of the CLPSO dynamic network, maybe it

**Table 19 DSWPSO with von Neumann neighborhood and two random neighbors.**

| | Fitness | | | Evaluations | | | |
|---|---|---|---|---|---|---|---|
| | Median | Min | Max | Median | Min | Max | SR |
| $f_1$ | 8.72e−12 | 1.07e−12 | 5.33e−11 | 20,188 | 18,767 | 22,589 | 50 |
| $f_2$ | 6.80E−36 | 5.61E−39 | 1.00e+04 | 151,704 | 121,765 | 218,393 | 49 |
| $f_3$ | 3.24e−11 | 1.14e−12 | 3.21e−10 | 22,981 | 20,972 | 26,166 | 50 |
| $f_4$ | 6.27e+01 | 2.69e+01 | 1.07e+02 | 11,417 | 5,586 | 31,654 | 47 |
| $f_5$ | 0.00e+00 | 0.00e+00 | 4.91e−02 | 19,477.5 | 17,101 | 25,627 | 50 |
| $f_6$ | 0.00e+00 | 0.00e+00 | 9.72e−03 | 7,448 | 2,989 | 28,567 | 43 |
| $f_7$ | 2.38e−02 | 0.00e+00 | 2.02e+00 | 34,937 | 32,977 | 40,180 | 20 |
| $f_8$ | 7.55e−15 | 4.00e−15 | 1.34e+00 | 20,972 | 18,767 | 24,892 | 47 |
| $f_9$ | 1.43e−05 | 6.42e−09 | 6.63e+03 | 639,842 | 374,066 | 901,110 | 41 |
| $f_{10}$ | 7.40e−03 | 0.00e+00 | 5.17e−01 | 21,021 | 18,130 | 25,284 | 47 |

**Table 20 DSWPSO with Moore neighborhood and two random neighbors.**

| | Fitness | | | Evaluations | | | |
|---|---|---|---|---|---|---|---|
| | Median | Min | Max | Median | Min | Max | SR |
| $f_1$ | 1.13e−12 | 8.12e−14 | 1.92e−11 | 19,306 | 17,395 | 21,119 | 50 |
| $f_2$ | 4.86e−38 | 2.52e−41 | 5.00e+03 | 141,708 | 121,079 | 219,520 | 45 |
| $f_3$ | 4.72e−12 | 7.08e−13 | 4.46e−11 | 22,050 | 19,845 | 25,480 | 50 |
| $f_4$ | 6.22e+01 | 3.48e+01 | 1.34e+02 | 10,731 | 6,958 | 23,520 | 47 |
| $f_5$ | 7.40e−03 | 0.00e+00 | 2.70e−02 | 18,497.5 | 16,611 | 20,531 | 50 |
| $f_6$ | 0.00e+00 | 0.00e+00 | 9.72e−03 | 6,811 | 3,136 | 25,480 | 48 |
| $f_7$ | 1.01e−01 | 0.00e+00 | 3.06e+00 | 35,035 | 32,683 | 39,494 | 16 |
| $f_8$ | 7.55e−15 | 4.00e−15 | 1.16e+00 | 20,090 | 16,954 | 24,941 | 47 |
| $f_9$ | 3.25e−05 | 4.29e−09 | 7.12e+03 | 620,487 | 365,981 | 916,692 | 35 |
| $f_{10}$ | 8.63e−03 | 0.00e+00 | 8.00e+00 | 19,747 | 17,052 | 25,235 | 43 |

is necessary to define a dynamic update strategy which takes into account the current set of particles from which an individual is learning at a specific period of the run. Steady-state updates strategies for PSO in dynamic networks is planned as future work.

## Dynamic small world PSO

The final experiment compares SS-PSO with the DSWPSO, recently proposed by *Vora & Mirlanalinee (2017)*. DSWPSO uses a static von Neumann topology to which a number of random connections are added in each iteration. It is a very simple variation of the standard PSO, but it attains quite interesting results when compared to a number of state-of-the-art PSOs.

For this paper, DSWPSO was tested with von Neumann and Moore topologies. The number of random neighbors in each topology was set to 2, as suggested by *Vora & Mirlanalinee (2017)*. Parameters $c_1$ and $c_2$ were set to 1.494 and ω to 0.7298. The algorithms were all run for 200,000 function evaluations. DSWPSO results are

**Table 21 Comparing SS-PSO and DSWPSO (von Neumann) with the Mann-Whitney test.**

| | $f_1$ | $f_2$ | $f_3$ | $f_4$ | $f_5$ | $f_6$ | $f_7$ | $f_8$ | $f_9$ | $f_{10}$ |
|---|---|---|---|---|---|---|---|---|---|---|
| Fitness | + | + | + | ≈ | ≈ | + | + | + | + | + |
| Eval. | + | + | + | + | + | + | + | + | ≈ | + |

**Notes:**
+If SS-PSO ranks first in the Mann–Whitney test and the result is significant.
−If DSWPSO ranks first and the results is significant.
≈If the differences are not significant.

**Table 22 Comparing SS-PSO and DSWPSO (Moore) with the Mann-Whitney test.**

| | $f_1$ | $f_2$ | $f_3$ | $f_4$ | $f_5$ | $f_6$ | $f_7$ | $f_8$ | $f_9$ | $f_{10}$ |
|---|---|---|---|---|---|---|---|---|---|---|
| Fitness | + | + | + | ≈ | ≈ | ≈ | + | + | + | + |
| Eval. | + | + | + | + | + | ≈ | + | + | ≈ | + |

**Notes:**
+If SS-PSO ranks first in the Mann–Whitney test and the result is significant.
−If DSWPSO ranks first and the results is significant.
≈If the differences are not significant.

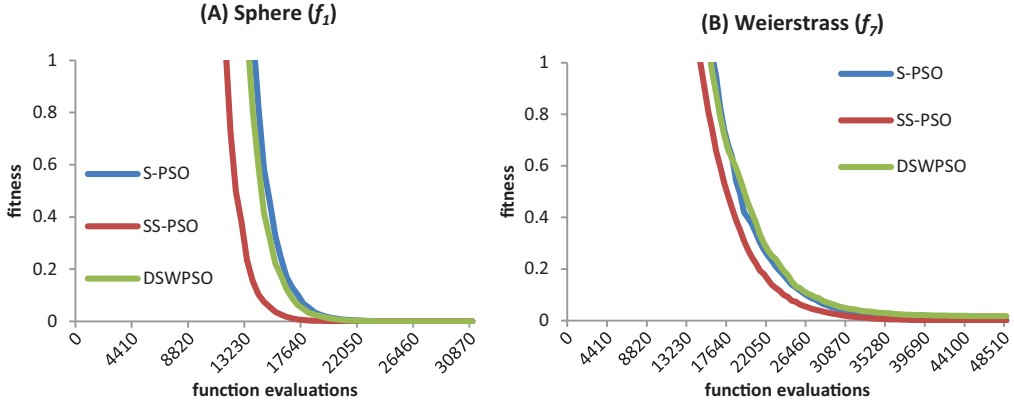

**Figure 10 S-PSO, SS-PSO and DSWPSO best fitness curves for the sphere (A) and Weierstrass (B) benchmark functions.**

presented in Table 19 (von Neumann) and Table 20 (Moore). The statistical analysis that compares SS-PSO and DSWPSO are in Table 21 (von Neumann) and Table 22 (Moore). It is clear that SS-PSO outperforms DSWPSO with both von Neumann and Moore base-topology in most of the functions, not only in terms of convergence speed, but also in solution quality.

Figure 10 shows the convergence curves (median best fitness values over 50 runs) of S-PSO, SS-PSO and DSWPSO (von Neumann). The graphics show that SS-PSO converges faster to the vicinity of the solutions. Furthermore, and although it is not perceivable in the graphics, SS-PSO eventually reaches solutions closer to $f(x) = 0$ (the optimum of both functions) as demonstrated by Tables 8 and 21.

## Running times

A final experiment compares S-PSO and SS-PSO running times. The algorithms are run on function $f_7$ with $D$ set to 10, 30, 50 and 100. Moore neighborhood is used in both

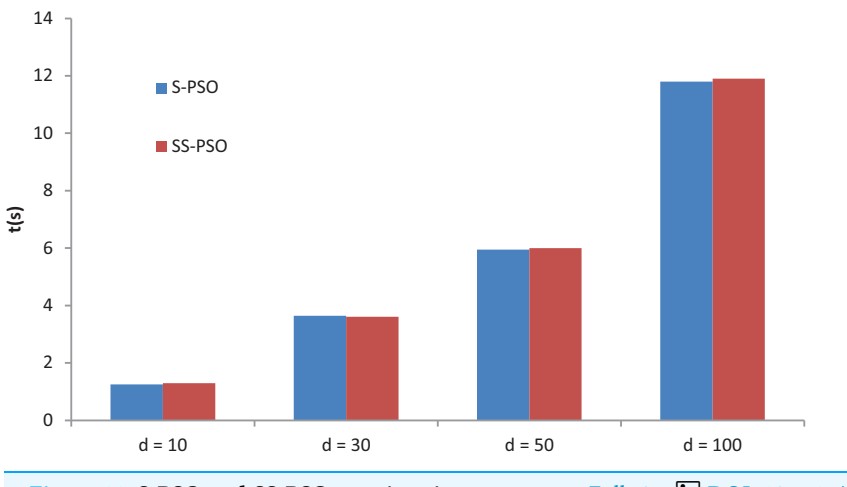

**Figure 11 S-PSO and SS-PSO running times.**

algorithms and parameters are set as in previous experiments. Figure 11 shows the running times of 49,000 functions evaluations (median values over 10 runs for each algorithm). The running times of each algorithm are statistically equivalent for every $D$ value. Running times of SS-PSO with von Neumann and Moore neighborhood are also equivalent. The PerfAndPubTools software (*Fachada et al., 2016*) was used to analyze the running times.

## DISCUSSION

The experiments in the previous sections demonstrate that SS-PSO is able to significantly improve the performance of the standard PSO, at least on the set of benchmark functions. The differences are particularly noticeable in the convergence speed of the algorithms, but SS-PSO is also able to improve the solution quality in several functions (see Table 8). An experiment comparing three different steady-state strategies show that replacing the worst particle and its neighbors is the best strategy. Our initial hypothesis (reducing the number of evaluations in each time step, while focusing only on the worst solutions, reduces the computational effort to reach a solution) is confirmed.

The relative performance of SS-PSO and standard PSO has also been verified for a wide range of parameter values (see Figs. 4–7) as well as for different problem dimensions (see Fig. 3). These results are important since they demonstrate that the proposed strategy has not been fine-tuned and that its validity is not restricted to a particular region of the parameter space or problem dimension. The algorithm was also compared to a PSO with time-varying acceleration, again attaining good results, thus reinforcing the idea that the steady-state strategy is consistent and robust. SS-PSO was compared to CLPSO, and while being outperformed in terms of solution quality in four functions, it yields better solutions in two problems, and is faster in other two functions. Since CLPSO is considered to be a very efficient algorithm, these results are promising. It deserves further examination whether variants of SS-PSO could clearly outperform CLPSO. Finally, SS-PSO was compared to DSWPSO with excellent results.

## CONCLUSIONS

This paper investigates the performance of a new and unconventional updated strategy for the PSO. The SS-PSO is inspired by the Bak–Sneppen model of coevolution. However, while in the Bak–Sneppen model the worst individual and its neighbors are replaced by random values, in SS-PSO the worst particle and its neighbors are updated and evaluated in each time step. The remaining particles are kept in a steady state until they eventually satisfy the update criterion. Due to its strategy, SS-PSO may be classified within the A-PSOs category. However, its working mechanisms are radically different from standard A-PSOs.

After preliminary tests that determined the best topology for a set of ten unimodal, multimodal, shifted, noisy and rotated benchmark problems, the strategy was implemented on the winning structure: two-dimensional lattice with Moore neighborhood. Quality of solutions, convergence speed and success rates were compared and statistical analyses were conducted on the results. SS-PSO significantly improved the performance of a standard S-PSO in every function, at least in one of the two criteria (quality of final solutions and convergence speed). A parameter sensitivity analysis showed that SS-PSO is not more sensitive to the variation of parameter values than S-PSO. A scalability test showed that the proposed strategy does not introduce scalability difficulties. The algorithm was compared to PSO-TVA, CLPSO and DSWPSO with good results.

The first step in future works is to increase the size of the test with more functions, hoping that an extended test set can improve our insight into the behavior of the algorithm. The emergent properties of the algorithm (size of events, duration of stasis, critical values) will be also studied and compared to those of the Bak–Sneppen model. Finally, steady-state update strategies in dynamic topologies will be investigated.

### Funding

This work was supported by Fundação para a Ciência e Tecnologia (FCT) Research Fellowship SFRH/BPD/66876/2009 and FCT Project (UID/EEA/50009/2013), EPHEMECH (TIN2014-56494-C4-3-P, Spanish Ministry of Economy and Competitivity), PROY-PP2015-06 (Plan Propio 2015 UGR), project CEI2015-MP-V17 of the Microprojects program 2015 from CEI BioTIC Granada. The funders had no role in study design, data collection and analysis, decision to publish, or preparation of the manuscript.

### Grant Disclosures

The following grant information was disclosed by the authors:
Fundação para a Ciência e Tecnologia (FCT), Research Fellowship: SFRH/BPD/66876/2009.
FCT PROJECT: UID/EEA/50009/2013.
EPHEMECH: TIN2014-56494-C4-3-P, Spanish Ministry of Economy and Competitivity.
PROY-PP2015-06: Plan Propio 2015 UGR.
CEI2015-MP-V17 of the Microprojects program 2015 from CEI BioTIC Granada.

## Competing Interests

The authors declare that they have no competing interests.

## Author Contributions

- Carlos M. Fernandes conceived and designed the experiments, performed the experiments, analyzed the data, contributed reagents/materials/analysis tools, prepared figures and/or tables, performed the computation work, authored or reviewed drafts of the paper, approved the final draft.
- Nuno Fachada conceived and designed the experiments, performed the experiments, analyzed the data, prepared figures and/or tables, performed the computation work, authored or reviewed drafts of the paper, approved the final draft.
- Juan-Julián Merelo authored or reviewed drafts of the paper, approved the final draft.
- Agostinho C. Rosa authored or reviewed drafts of the paper, approved the final draft.

## Data Availability

Data is available at GitHub: https://github.com/laseeb/openpso.

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
