# Peer review of "Steady state particle swarm"

_PeerJ Computer Science, doi:10.7717/peerj-cs.202_

## Round 0.1 · original submission · Major Revisions

While the algorithm describes a potentially interesting new PSO variant, the reviewers raise valid concerns about how it is described and how results are reported among other things. When it comes to parallelization, the authors need to either improve the presentation substantially or to cut that part out and make it another paper.

Reviewer 1 ·

Basic reporting

This paper proposed a steady state update strategy for PSO algorithms. The proposed algorithm was tested on a set of unimodal, multimodal, noisy and rotated benchmark functions. Finally, a parallel SS-PSO has been implemented and compared to the standard PSO. But some weakness can be observed:
(1) The latest references should be cited. The review of the existing PSO algorithms is very necessary.

Experimental design

(3) The results are not persuasive. Comparison with the standard PSO is not acceptable, the authors should present their results compare to related papers. Is the proposed algorithm better than other swarm algorithms?
(4) The 'Mean' and 'Std' values should be provided in Table 2.
(5) The convergence curves of the proposed algorithm and the other algorithms should be added.

Validity of the findings

The steady state update strategy for PSO algorithm is not descrided clearly enough in the paper. So the paper is lack of innovation.

Additional comments

This paper proposed a steady state update strategy for PSO algorithms. The proposed algorithm was tested on a set of unimodal, multimodal, noisy and rotated benchmark functions. Finally, a parallel SS-PSO has been implemented and compared to the standard PSO. But some weakness can be observed:
(1) The steady state update strategy for PSO algorithm is not descrided clearly enough in the paper. So the paper is lack of innovation.
(2) The latest references should be cited. The review of the existing PSO algorithms is very necessary.
(3) The results are not persuasive. Comparison with the standard PSO is not acceptable, the authors should present their results compare to related papers. Is the proposed algorithm better than other swarm algorithms?
(4) The 'Mean' and 'Std' values should be provided in Table 2.
(5) The convergence curves of the proposed algorithm and the other algorithms should be added.

Cite this review as

Reviewer 2 ·

Basic reporting

This paper address an interesting topic: A Steady State OpenMp parallel version of PSO.

Authors take inspiration from SOC as applied by Bak-Sneppen to coevolving species in 1993. Authors are not only interested in the model, but also in the possibilities for parallel implementations, and their benefits.

The topic address is of interest, and the paper is well-written. However, it is not clear to me why the "parallel" part of the study has been included: it is a small section of the paper, that would really require a large study and probably an additional paper. Actually, when I read the title, given the inclusion of "OpenMP parallel" I though this paper would really deal with a parallel algorithm, which is not the case. Right now it is confusing what the main goal of the paper is.

If we focus on parallel versions of PSO, authors describe the problem of unbalanced distribution of particles when synchronous models are used. In other areas, load-balancing is a topic always addressed, and a wider discussion is pertinent here, given that methods and algorithms have been widely studied and are available in this area (76-80). So, the literature review should specifically address this topic.

Authors state that quality of solutions and speed are the main goals, and the proposal relies in reducing the number of evaluations every time step. This is not a new idea, and has been frequently used in the context of Eas reducing fitness cases evaluations, when every individual must check a number of cases. (88-90) If this is one of the main ideas here, a review of the literature would be pertinent so that readers can understand the connection with previously published papers on the topic.

On the other hand, steady state approaches are not new for population-based metaheuristics, and are also frequent in the EA literature. Nevertheless, the traditional approach is that a single population is managed, where new individuals are born, and only they are evaluated every time step, and some pass over, so that the size of the population does not change. A comparison among the approach described and the standard one should be included in the review, providing references to interested readers. (100).

Experimental design

(194) A description on how the algorithms has been tested should be included: although that information is provided below (50 runs per experiment), this is the best place to include it.

Given that two particular values of neighbourhoods has been tested, (5 and 9), and given that each of them require different computing time, results should also include time to solution for a better assessment of results. A wider study should include the range of neighbourhoods values so that we can understand why some values are better than others. If this is one of the main components of the algorithm, better to test them in the context of the new proposal.


(283) some comments should be added about why in f1 and f3 the new proposal does not found better results.

(402) the first and only time when references to tables are correctly included. Why this time are correct and previously never? (error in other cases).

Validity of the findings

Although results shown for the algorithm are clear, regarding the parallelization section there are some problems that must be addressed.

(416) Parallelization

The topic is important enough to be addressed separately. But the information provided is quite scarce to be able to understand the implications. For instance, no information on how the parallelization has been applied is included. Authors state that run-time has been compared when using 1-8 threads. But what technology, programming library, etc, for parallelising the code has been applied? Are authors simply relying on compilers decisions? Even if the code is shared, the paper should include the required information for an occasional reader to know the details without checking the code.

(434) authors refers here to parallelisation at the evaluation level. But the results in this case should provide fitness quality against time for a proper comparison. Speedup is not enough. Also comparison between time-to-solution should be included.

(441) Authors state: “Unfortunately, large run time variations were observed, making it difficult to perform a direct comparison. This was to be expected, since a single unsuccessful run is enough to skew the results towards either algorithm. Repeating the test with different seeds often lead to opposing results.”

This shouldn't avoid a correct comparison: maybe number of successful runs under time intervals could be used. Also, time provided can be longer thus assuring all the runs finish; or difficulty of problems increased, so that no run finish along the time studied, and quality can be analyzed along the experiment.

Given the specific problems that arise when parallelisation is addressed, I suggest to remove this topic from the paper and try it as a specific topic in a future paper.

Additional comments

I don't understand why no correct links to figures and tables are included in the paper, and instead only “Error” messages are included.

Cite this review as

---

## Round 0.2 · accepted · Accept

I, and the reviewer, are happy with your revisions - I think this is ready to be published now.

Reviewer 1 ·

Basic reporting

no comment

Experimental design

no comment

Validity of the findings

no comment

Additional comments

no comment

Cite this review as